# An Ensemble-Based Statistical Methodology to Detect Differences in Weather and Climate Model Executables

Christian Zeman[1] and Christoph Schär[1]

[1]Institute for Atmospheric and Climate Science, ETH Zurich, Switzerland

**Correspondence:** Christian Zeman (christian.zeman@env.ethz.ch)

**Abstract.** Since their first operational application in the 1950s, atmospheric numerical models have become essential tools in weather prediction and climate research. As such, they are subject to continuous changes, thanks to advances in computer systems, numerical methods, more and better observations, and the ever-increasing knowledge about the atmosphere of Earth. Many of the changes in today's models relate to seemingly innocuous modifications associated with minor code rearrangements, changes in hardware infrastructure, or software updates. Such changes are meant to preserve the model formulation, yet the verification of such changes is challenged by the chaotic nature of our atmosphere – any small change, even rounding errors, can have a significant impact on individual simulations. Overall this represents a serious challenge to a consistent model development and maintenance framework.

Here we propose a new methodology for quantifying and verifying the impacts of minor atmospheric model changes or its underlying hardware/software system by using ensemble simulations in combination with a statistical hypothesis test for instantaneous or hourly values of output variables at a grid-cell level. The methodology can assess the effects of model changes on almost any output variable over time and be used with different underlying statistical hypothesis tests.

We present first applications of the methodology with the regional weather and climate model COSMO. While providing very robust results, the methodology shows a great sensitivity even to very small changes. Specific changes considered include applying a tiny amount of explicit diffusion, the switch from double- to single-precision, and a major system update of the underlying supercomputer. Results show that changes are often only detectable during the first hours, suggesting that short-term ensemble simulations (days to months) are best suited for the methodology, even when addressing long-term climate simulations. Furthermore, we show that spatial averaging – as opposed to testing at all grid points – reduces the test's sensitivity for small-scale features such as diffusion. We also show that the choice of the underlying statistical hypothesis test is not essential and that the methodology already works well for coarse resolutions, making it computationally inexpensive and therefore an ideal candidate for automated testing.

## 1 Introduction

Today's weather and climate predictions heavily rely on data produced by atmospheric numerical models. Ever since their first operational application in the 1950s, the models have been improved thanks to advances in computer systems, numerical methods, observational data, and the understanding of the Earth's atmosphere. While such changes often may be only small

and incremental, accumulated they have a big effect which manifests itself in a significant increase in skill of weather and climate predictions over the past 40 years (Bauer et al., 2015).

While some of the model changes are intended to extend and improve the model, others are not meant to affect the model results but merely its computational performance and versatility. In software engineering, one often distinguishes between "upgrades" and "updates" in such cases. For weather and climate models, an upgrade would, for example, be the introduction of a new and improved soil model, whereas a new version of underlying software or a binary that has been built with a newer compiler version would only represent an update. Updates are often employed due to the necessity of keeping the software up-to-date without any perceivable improvements in functionality. For a weather and climate model, the model results are not supposed to be significantly affected by such an update. This also applies to other changes, such as moving to a different hardware architecture or changing the domain decomposition for distributed computing. A robust behavior of the model with regard to such changes is crucial for a consistent interpretation of the results and the credibility of the derived predictions and findings.

Weather and climate model results are generally not bit-identical when they are, for example, run on different hardware architectures or have been compiled with a different compiler. This is because the associativity property does not hold for floating-point operations (i.e., $(x + y) + z = x + (y + z)$ is not given), and the fact that the order of arithmetic operations is dependent on the compiler and the targeted hardware architecture. Schär et al. (2020) have achieved bit-reproducibility for the regional weather and climate model COSMO between a CPU and a GPU version of the model by limiting instruction rearrangements from the compiler and with the use of a preprocessor that automatically adds parentheses to every mathematical expression of the model. However, this also came with a performance penalty where the CPU and GPU bit-reproducible versions were slower by 37% and 13%, respectively, than their non-bit-reproducible counterparts. Due to this performance penalty and the effort involved in making a model bit-reproducible, bit-reproducibility is generally not enforced. It has to be noted that this behavior of not producing bit-identical results across different architectures or when using different compilers is common for most computer applications and not a problem per se. However, for weather and climate models, it represents a serious challenge due to the chaotic nature of the underlying nonlinear dynamics, where small changes can have a big effect (Lorenz, 1963). For example, a tiny difference in the initial conditions of a weather forecast can potentially lead to a very different prediction. Consequently, also rounding errors can affect the model results in a major way. In order to mitigate this effect and to provide probabilistic predictions, forecasts often use ensemble prediction systems (EPS), where a model is run several times for the same time frame with slightly perturbed initial conditions or stochastic perturbations of the model simulations (see Leutbecher and Palmer, 2008, for an overview). The use of an EPS accounts for the uncertainty in initial conditions and the internal variability of the model results.

So in order to verify whether the properties of a weather and climate model executable are not significantly affected after an update or a change to a different platform, we have to resort to ensemble simulations. Without ensemble simulations, we would only be able to answer something we already know a priori: Any change in the model or its underlying software and hardware will make the model slightly different and, therefore, might significantly affect the output due to the chaotic nature of the underlying dynamics. However, with ensemble simulations, we can answer the much more important question: How do

the changes of model results compare to the internal variability of the underlying nonlinear dynamical system? If the effect of the new model is significantly smaller than the one of internal variability, a statistical test will not be able to detect whether the results of the new and the old model come from the same distribution or not.

In this paper, the detection of such changes will be referred to as "verification". In the atmospheric and climate science community, the terms "validation" and "verification" are not always used in a clearly defined way and sometimes even used interchangeably. An extensive discussion about different definitions of verification and validation can be found in Oberkampf and Roy (2010). Sargent (2013) defines verification as "ensuring that the computer program of the computerized model and its implementation are correct". In contrast, validation is defined as "substantiation that a model within its domain of applicability possesses a satisfactory range of accuracy consistent with the intended application of the model". According to Carson (2002), validation refers to "the processes and techniques that the model developer, model customer and decision makers jointly use to assure that the model represents the real system (or proposed real system) to a sufficient level of accuracy", while verification refers to "the processes and techniques that the model developer uses to assure that his or her model is correct and matches any agreed-upon specifications and assumptions". Clune and Rood (2011) define validation as "comparison with observations" and verification as "comparison with analytic test cases and computational products". Whitner and Balci (1989) state that "whenever a model or model component is compared with reality, validation is performed", whereas they define verification as "substantiating that a simulation model is translated from one form into another, during its development life cycle, with sufficient accuracy". Oreskes et al. (1994) and Oreskes (1998) recommend not to use the terms verification and validation for models of complex natural systems at all. They argue that both terms imply an either-or situation for something that is not possible (i.e., a model will never be able to accurately represent the actual processes occurring in a real system) or only possible to evaluate for simplified and limited test cases (i.e., comparing with analytical solutions for simple problems). Nevertheless, both terms are commonly used in atmospheric sciences. Note that in this paper, we follow the terminology of Whitner and Balci (1989). As our methodology's goal is to ensure that there are no significant differences between two model executables, we use the term verification for the methodology.

Using the definition from Whitner and Balci (1989), verification is a form of system testing in the area of software engineering. This means that a complete integrated system is tested, in this case, a weather and climate model consisting of many different components that interact with each other. System tests are an integral part of testing in software engineering. An objective system test that can be performed automatically is also an excellent asset for the practice of continuous integration and continuous deployment (CI/CD). CI/CD enforces automation in building, testing and deployment of applications and should also be considered good practice in developing and operating weather and climate models.

## 2 Background

### 2.1 Current state of the art

Despite its importance for the consistency and trustworthiness of model results, verification has received relatively little attention in the weather and climate community. However, the awareness seems to have increased, as some recent studies tackle this issue more systematically.

Rosinski and Williamson (1997) were among the first to propose a strategy for verifying atmospheric models after they had been ported to a new architecture. They set the conditions that the differences should be within one or two orders of magnitude of machine rounding during the first few time steps and that the growth of differences should not exceed the growth of initial perturbations at machine precision during the first few days. The methodology of Rosinski and Williamson (1997) was developed and used for the NCAR Community Climate Model (CCM2). However, the approach is no longer applicable for its current successor, the Community Atmosphere Model (CAM), because the parameterizations are ill-conditioned, which makes small perturbations grow very quickly and exceed the tolerances of rounding error growth within the first few timesteps (Baker et al., 2015). Thomas et al. (2002) performed 42-hour simulations with the Mesoscale Compressible Community (MC2) model to determine the importance of processor configuration (domain decomposition), floating-point precision, and mathematics libraries for the model results. By analyzing the spread of runs with different settings, they concluded that processor configuration is the main contributor among these categories to differences in the results of their dynamical core. Knight et al. (2007) analyzed an ensemble of over 57,000 climate runs from the climateprediction.net project (www.climateprediction.net, last access: 31 January 2022). The climate runs have been performed with varying parameter settings and initial conditions on different hardware and software architectures. Using regression tree analysis, they demonstrated that the effect of hardware and software is small relative to the effects of parameter variations and, over the wide range of systems tested, may be treated as equivalent to that caused by changes in initial conditions. Hong et al. (2013) performed seasonal simulations with the global model program (GMP) of the Global/Regional Integrated Model system (GRIMs) on ten different software system platforms with different compilers, parallel libraries, and optimization levels. The results showed that the ensemble spread caused by differences in the software system is comparable to that caused by differences in initial conditions.

One of the most comprehensive recent studies on verification is from Baker et al. (2015), where they proposed the use of principal component analysis (PCA) for consistency testing of climate models. Instead of testing all model output variables, many highly correlated, they only looked at the first few principal components of the model output and used z-scores to test if the value from a test configuration is within a certain number of standard deviations from the control ensemble. If the test failed for too many PCs, they rejected the new configuration. They confirmed their methodology using 1-year long simulations of the Community Earth System Model (CESM) with different parameter settings, hardware architectures, and compiler options. While the methodology showed high sensitivity and promising results, it had some difficulties detecting changes caused by additional diffusion due to its focus on annual global mean values. Baker et al. (2016) also used z-scores for consistency testing of the Parallel Ocean Program (POP), the ocean model component of the Community Earth System Model (CESM). However, instead of evaluating principal components on spatial averages, as in Baker et al. (2015), they applied the methodology at

each grid point for individual variables and stipulated that this local test has to pass for at least 90% of the grid points to have the global test pass. Milroy et al. (2018) extended the consistency test by Baker et al. (2015) by performing the test on spatial means for the first nine time steps of the Community Atmospheric Model (CAM) on a global 1° grid with a time step of 1800 s. With this method, they were able to produce the same results for the same test cases as Baker et al. (2015). Additionally, they were also able to detect small changes in diffusion, which were not detected in Baker et al. (2015).

Wan et al. (2017) used time-step convergence as a criterion for model verification, based on the idea that a significantly different model executable will no longer converge towards a reference solution produced with the old executable. Their test methodology produced similar results as the one from Baker et al. (2015) and is relatively inexpensive due to the short integration times. However, due to the nature of the test, it cannot detect issues associated with diagnostic calculations that do not feedback to the model state variables.

Mahajan et al. (2017) used an ensemble-based approach where they applied the Kolmogorov-Smirnov (K-S) test on annual and spatial means of 1-year simulations for testing the equality of distributions of different model simulations. Furthermore, they used generalized extreme value (GEV) theory for representing the annual maxima of daily average surface temperature and precipitation rate. They then applied a Student's t-test on the estimated GEV parameters at each grid-point to test the occurrence of climate extremes. They showed that the climate extremes test based on GEV theory was considerably less sensitive to changes in optimization strategies than the K-S test on mean values. Mahajan et al. (2019) applied two multivariate two-sample equality of distribution tests, the energy test and the kernel test, on year-long ensemble simulations following Baker et al. (2015) and Mahajan et al. (2017). However, both these tests generally showed a lower power than the K-S test from Mahajan et al. (2017), which means that more ensemble members were needed to reject the null hypothesis confidently. Mahajan (2021) used the K-S test as well as the Cucconi test for annual mean values at each grid point for the verification of the ocean model component of the US Department of Energy's Energy Exascale Earth System Model (E3SM). Furthermore, they used the False Discovery Rate (FDR) method by Benjamini and Hochberg (1995) for controlling the false positive rate. Both tests were able to detect very small changes of a tuning parameter, with the K-S test showing a slightly higher power than the Cucconi test for the smallest changes.

Massonnet et al. (2020) recently proposed an ensemble-based methodology based on monthly averages (and an average over the whole simulation time), followed by the comparison of these averages on a grid-cell level against standard indices used in Reichler and Kim (2008). Finally, spatially averaging results in one scalar number per field, month, and ensemble member. These scalars were then used for the K-S test to detect statistically significant differences. Performing this test for climate runs with the EC-Earth earth system model version 3.1 on different computing environments revealed significant differences for 4 out of 13 variables. However, the same test for the newer EC-Earth 3.2 version showed no significant differences. Massonnet et al. (2020) suspect the presence of a bug in EC-Earth 3.1 and its subsequent fix for version 3.2 as the reason for this disparity.

## 2.2 Determining field significance

A challenging question in the area of model verification is the role of statistical significance at the grid-point versus the field level. A statistical hypothesis test's significance level $\alpha$ is defined as the probability of rejecting the null hypothesis even though

the null hypothesis is true (commonly known as false positive or type I error). So if we compare two ensembles and perform the test at every grid point, the test may locally reject the null hypothesis even if the two ensembles stem from the same model.

When assuming spatial independence, the probability of having $x$ rejected local null hypotheses out of $N$ tests follows from the binomial distribution:

$$P(x) = \frac{N!}{x!(N-x)!} \alpha^x (1-\alpha)^{N-x} \tag{1}$$

On average, we can expect $\alpha N$ local rejections over the whole grid when two ensembles come from the same model. However, for $N = 100$ and $\alpha = 0.05$, the probability of having 9 or more erroneous rejections is still 6.3%, which means that 10 or more local rejections are required (probability 2.8%) to reject the global null hypothesis on field level with a 95% confidence interval. So, in this case, 10% of the local hypothesis tests would have to reject the local null hypothesis to get a significant global rejection. For a larger grid with $N = 10000$, we would require 537 (5.37%) or more local rejections (probability 4.8%) to reject the global null hypothesis with a 95% confidence interval (see Fig. 3 in Livezey and Chen, 1983, for a visualization of this function).

However, local tests cannot be assumed to be statistically independent due to spatial correlation. Therefore, Eq. (1) is not valid in our case. While two identical models will still have $\alpha N$ false rejections on average, a higher or lower rejection rate is more likely. Unfortunately, the exact distribution of rejection rates is unknown in such a case (Storch, 1982). Livezey and Chen (1983) argued that spatial correlation reduces $N$, the number of independent tests, due to a clustering effect of grid points and therefore also increases the percentage of local rejections needed to reject the global null hypothesis. To account for that, they estimated the effective number of independent tests $N_{\text{eff}}$ with the use of Monte Carlo methods, which allowed them to use Eq. (1) for calculating the number of rejected local tests that are required to reject the global null hypothesis.

Wilks (2016) recommended the use of the False Discovery Rate (FDR) method by Benjamini and Hochberg (1995). This method defines a threshold level $p_{\text{FDR}}$, based on the sorted p-values. The threshold is defined as

$$p_{\text{FDR}} = \max_{i=1,\ldots,N} \left[ p_{(i)} : p_{(i)} \leq (i/N)\alpha_{\text{FDR}} \right] , \tag{2}$$

where $p_{(i)}$ are the sorted p-values with $i = 1, \ldots, N$ and $\alpha_{\text{FDR}}$ is the chosen control level for the FDR (note that $\alpha_{\text{FDR}}$ must not be the same as $\alpha$ for the local test). The FDR method only rejects local null hypotheses if the respective p-value is no larger than $p_{\text{FDR}}$. This condition essentially ensures that the fraction of false rejections out of all rejections is at most $\alpha_{\text{FDR}}$ on average. While the FDR method by Benjamini and Hochberg (1995) is theoretically also based on the assumption that the different tests are statistically independent, it has been shown to also effectively control the proportion of falsely rejected null hypotheses for spatially correlated data (Ventura et al., 2004; Mahajan, 2021). An assessment of the FDR method in the context of our verification methodology will be presented in Sect. 4.11.

## 3 Methods and data

### 3.1 Verification methodology

We consider ensemble simulations of two model versions, which for brevity will be referred to as "old" and "new", respectively. We start by stating our global null hypothesis:

$H_{0\,(\mathrm{global})}$: The ensemble results from the old and the new model are drawn from the same distribution.

We then consider the changes in the model to be insignificant if we are not able to reject the global null hypothesis. This global test is based on a statistical hypothesis test applied on a grid-cell level with a local null hypothesis $H_{0\,(i,j)}$. The specific definition of $H_{0\,(i,j)}$ will be given later, as it somewhat depends upon the chosen statistical hypothesis test; see Sect. 3.3. It is also important to state that we will generally not evaluate the whole model output but compare a limited number of two-dimensional fields, such as the 500 hPa geopotential height or the 850 hPa temperature fields. For each selected field, the two model ensembles will be tested at grid-scale against each other, using an appropriate statistical test. The probability of rejecting $H_{0\,(i,j)}$ for two ensembles produced by an identical model is given by the significance level $\alpha$ (here, $\alpha = 0.05$). As discussed in Sect. 2.2, the main difficulty of using statistical hypothesis tests on a grid-cell level is the spatial correlation, making the respective tests not statistically independent and thus prohibiting the use of the binomial distribution for calculating the probabilities of false positives. We chose to deal with this in a conceptually simple but effective way. The methodology follows Livezey (1985) and combines Monte Carlo methods and subsampling to produce a null distribution of rejection rates, which can be used to get the probability of having $n_{\mathrm{rej}}$ rejections for two ensembles coming from the same model. An alternative to generating the null distribution from a control ensemble is the use of Monte Carlo permutation testing, where one pools two ensembles (from which one does not know yet whether they come from the same distribution), and then applies the test to randomly drawn subsets from the pooled ensemble. This approach allows bypassing the creation of a control ensemble and therefore save compute time. Strictly speaking, the reference value for the number of rejections then comes from a distribution not produced by one but by two models. Depending on the difference between the two models, this might lead to slightly different results compared to a case where the reference distribution comes from two identical models. However, Mahajan et al. (2017) and Mahajan et al. (2019) used both approaches and found only minor differences between permutation testing and subsampling from a control ensemble to generate the null distribution. Nevertheless, we still opted for the approach with a control ensemble since the additionally needed compute time is relatively small for short simulations (see Sect. 3.5).

Figure 1 shows a schematic example of the procedure. The control and reference ensembles come from an identical model (old model), whereas the evaluation ensemble comes from a model where we are unsure whether it produces statistically indistinguishable results (new model). Each ensemble consists of $n_E$ members and we use $m$ subsamples consisting of $n_S$ random members ($n_S < n_E$) drawn from each ensemble without replacement. We then test for field significance by comparing the mean rejection rate from the evaluation ensemble to the 0.95 quantile from the control ensemble, rejecting the null hypothesis if the mean rejection rate of the evaluation ensemble is equal to or above the 0.95 quantile of the control ensemble rejection rate.

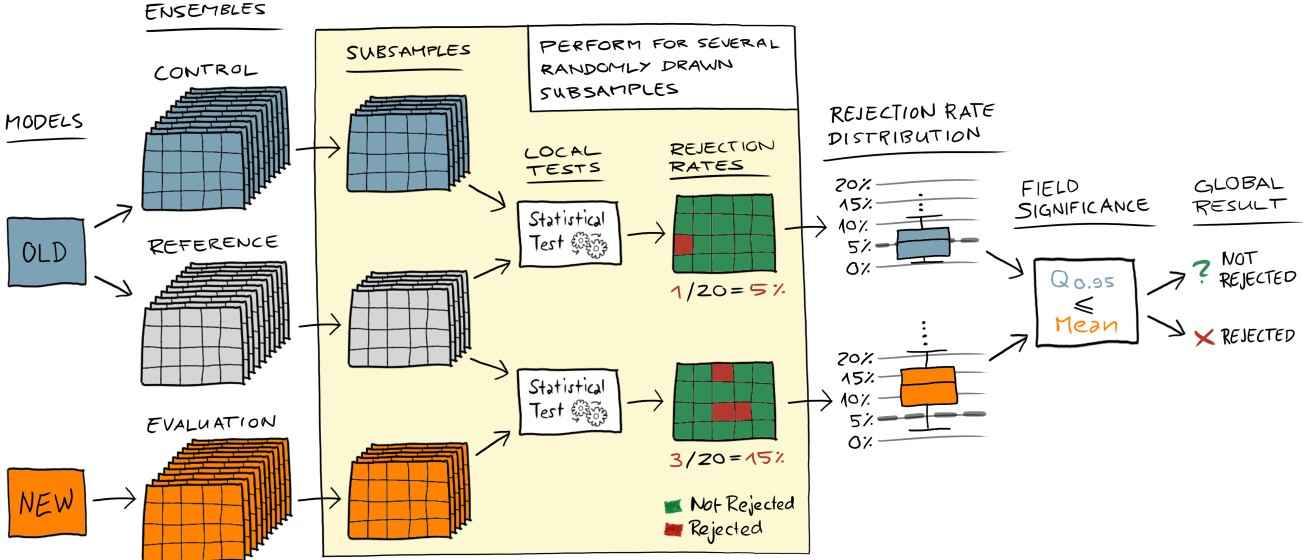

**Figure 1.** Schematic sketch of the verification methodology. The control and the reference ensemble come from the same "old" model, whereas the evaluation ensemble comes from a "new" model, where we do not know whether it is indistinguishable from the model that created the control and reference ensemble. We draw many random subsamples from all three ensembles, perform the local statistical hypothesis tests of the control and evaluation subsamples against the reference subsamples, and then calculate the rejection rate for each subsample. This results in a distribution of the rejection rates for the control and evaluation ensemble, which can then be compared to each other in order to decide whether the evaluation ensemble is different. In this work, we reject the global null hypothesis if the mean of the evaluation ensemble rejection rate distribution is equal to or above the 0.95 quantile of the rejection rate distribution from the control ensemble.

Next to accounting for spatial correlation, having a rejection rate distribution from a control ensemble also offers more flexibility in evaluating different variables. In atmospheric models, some variables, such as precipitation, inherently have a high probability of zero values at many grid points. Therefore, a statistical test will often not reject the local null hypothesis even though the two ensembles might come from two very different models. This can lead to a mean rejection rate well below $\alpha$ for two different ensembles, and by just looking at $\alpha$, we would conclude that the two ensembles are indistinguishable. However, here we derive the expected rejection rate from the control ensemble, which yields an objective threshold that accounts for such behavior.

It is important to mention that the choice of $\alpha = 0.05$ for the local statistical hypothesis test is arbitrary and does not determine the confidence interval for field significance. Furthermore, comparing the mean rejection rate from the evaluation ensemble with the 0.95 quantile from the control might also give a wrong idea of a confidence interval for the field significance. If we assume that the evaluation ensemble comes from an identical model and only take one subsample from the evaluation ensemble, the probability of it having a rejection rate equal to or higher than the 0.95 quantile from the control rejection rate

distribution is, in fact, 5%. However, the probability of the mean rejection rate of 100 subsamples from the evaluation ensemble being higher than the 0.95 quantile of the control is significantly lower than 5%, but it is not easy to determine by how much. Using the binomial distribution in Eq. (1) for a calculation of the number of necessarily rejected subsamples to reject the overall null hypothesis is not valid, because the subsamples are not statistically independent from each other. Based on our experience and the results shown in this work, we consider the comparison of the mean to the 0.95 quantile a reasonable choice, even though it is not really based on a confidence interval (unlike, for example, the FDR approach discussed in Sect. 2.2). However the sensitivity of the methodology could of course be adapted by changing this field significance criterion.

The verification methodology in this work shares some similarities with verification methodologies presented in previous studies, most notably Baker et al. (2015, 2016); Milroy et al. (2018); Mahajan et al. (2017, 2019); Mahajan (2021); Massonnet et al. (2020). However, most of these studies focus on mean values in space and time. From the previously mentioned studies, only Baker et al. (2016), Mahajan et al. (2017), and Mahajan (2021) have used a similar methodology on a grid cell level, either for monthly or yearly averages of variables from an ocean model component (Baker et al., 2016; Mahajan, 2021) or for the identification of differences in annual extreme values (Mahajan et al., 2017). Moreover, except for Milroy et al. (2018), all other studies focus on longer simulations (one year or more) and average values in time. We will focus on shorter simulations (days to months) with the idea that many small changes are often easier to identify at the beginning of the simulations. We apply the methodology directly to instantaneous or, in the case of precipitation, hourly output variables from an atmospheric model on a 3-hourly or 6-hourly basis. The rejection threshold is computed as a function of time and may transiently increase or decrease in response to changes in predictability. In essence, the rejection rate distribution from a control ensemble allows us to use an objective criterion for field significance. Another difference to most existing verification methodologies is that this methodology calculates the mean rejection rate from the evaluation ensemble and the 0.95 quantile from the control ensemble using subsampling. It thus essentially performs multiple global tests to arrive at a pass or fail decision. Most existing methodologies use only one test with all ensemble members for the pass or fail decision. However, many of them use subsampling to estimate the false positive rate.

## 3.2 Ensemble generation

The ensemble is created through a perturbation of the initial conditions of the prognostic variables (in our case, horizontal and vertical wind components, pressure perturbation, temperature, specific humidity, and cloud water content). The perturbed variable $\hat{\varphi}$ is defined as

$$\hat{\varphi} = (1 + \epsilon R)\varphi, \tag{3}$$

where $\varphi$ is the unperturbed prognostic variable, $R$ a random number with a uniform distribution between $-1$ and $1$, and $\epsilon$ the specified magnitude of the perturbation. In this study, we have used $\epsilon = 10^{-4}$ for all experiments. Next to providing a good ensemble spread already during the first few hours, the relatively strong perturbation also works well with single-precision floating-point representation. Furthermore, the effect on internal variability with $\epsilon = 10^{-4}$ is very similar to the one from much weaker perturbations (e.g., $\epsilon = 10^{-16}$) after a few hours, as shown in Appendix A.

## 3.3 Statistical hypothesis tests

In this study, we have applied three different statistical tests for testing the local null hypothesis $H_{0(i,j)}$: the Student's t-test, the Mann-Whitney U (MWU) test, and the two-sample Kolmogorov-Smirnov (K-S) test. This allows us to see whether some statistical tests might be better suited for some variables than others and how sensitive the methodology is with regard to the underlying test statistics. If not mentioned otherwise, the MWU test has been used as the default test for the results shown in 270 this study.

### 3.3.1 Student's t-test

The Student's t-test was introduced by William S. Gosset under the pseudonym "Student" (Student, 1908) and has been originally used to determine the quality of raw material of stout for the Guinness Brewery. The independent two-sample t-test has the null hypothesis that the means of two populations $X$ and $Y$ are equal. As we use it for the local statistical test, we therefore 275 have the following local null hypothesis:

$H_{0(i,j)}$:   The means $\overline{\varphi_{\text{old}(i,j)}}$ and $\overline{\varphi_{\text{new}(i,j)}}$ are drawn from the same distribution.

Here, $\overline{\varphi_{\text{old}(i,j)}}$ is the sample mean of the variable $\varphi$ at grid cell $(i,j)$ from the old model, and $\overline{\varphi_{\text{new}(i,j)}}$ is the respective sample mean from the new model. The t statistic is calculated as

$$t = \frac{\overline{X} - \overline{Y}}{s_p \sqrt{\frac{2}{n_S}}}, \tag{4}$$

with $\overline{X}$ and $\overline{Y}$ being the respective sample means and assuming equal sample size $n_S = n_X = n_Y$. The pooled standard deviation is given as

$$s_p = \sqrt{\frac{s_X^2 + s_Y^2}{2}}, \tag{5}$$

where $s_X^2$ and $s_Y^2$ are the unbiased estimators of the variances of the two samples. The t statistic is then compared against a critical value for a certain significance level $\alpha$ from the Student's t-distribution. For a two-sided test, we reject the local null 285 hypothesis if the t statistic is smaller or greater than this critical value. The Student's t-test requires that the means of the two populations should follow a normal distribution and assumes equal variance. However, the Student's t-test has been shown to be quite robust to violations of both the normality assumption and, provided the sample sizes are equal, the assumption of equal variance (Bartlett, 1935; Posten, 1984). Sullivan and D'Agostino (1992) showed that the Student's t-test even provided meaningful results in the presence of floor effects of the distribution (i.e., where a value can be at minimum zero).

### 3.3.2 Mann-Whitney U test

The Mann-Whitney U (MWU) test (also known as Wilcoxon rank-sum test) has been introduced by Mann and Whitney (1947) and is a non-parametric test, in the sense that no assumption is made concerning the distribution of the variables. The null

hypothesis is that for randomly selected values $X_k$ and $Y_l$ from two populations, the probability of $X_k$ being greater than $Y_l$ is equal to the probability of $Y_l$ being greater than $X_k$. It therefore does not test exactly the same property as the Student's t-test (means of two populations are equal), even though it is often compared to it. In our case, the local null hypothesis test for the MWU test is the following:

$H_{0\,(i,j)}$: The probability of $\varphi^k_{\text{old}(i,j)} > \varphi^l_{\text{new}(i,j)}$ is equal to the probability of $\varphi^k_{\text{old}(i,j)} < \varphi^l_{\text{new}(i,j)}$.

Here, $\varphi^k_{\text{old}(i,j)}$ and $\varphi^l_{\text{new}(i,j)}$ are the values of the variable $\varphi$ at location $(i,j)$ from randomly selected members $k$ and $l$ of the samples from the old and new model respectively. The MWU test ranks all the observations (from both samples combined in one set) and then sums up the ranks of the observations from the respective samples, resulting in $R_X$ and $R_Y$. $U_{\min}$ is calculated as

$$U_{\min} = \min\left( R_X - \frac{n_X(n_X+1)}{2}, R_Y - \frac{n_Y(n_Y+1)}{2} \right),\tag{6}$$

where $n_X$ and $n_Y$ are the respective sample sizes, which are assumed to be equal in our case ($n_X = n_Y = n_S$). This value is then compared with a critical value $U_{\text{crit}}$ from a table for a given significance level $\alpha$. For larger samples ($n_S > 20$), $U_{\text{crit}}$ is assumed to be normally distributed. If $U_{\min} \leq U_{\text{crit}}$ the null hypothesis is rejected. As a non-parametric test, the MWU test has no strong assumptions and just requires the responses to be ordinal (i.e., $<, =, >$). Zimmerman (1987) showed that, given equal sample sizes, the MWU test is a bit less powerful than the Student's t-test, even if variances are not equal. This means that the probability of correctly rejecting the null hypothesis, when the alternative hypothesis is true, is assumed to be a bit lower. Nevertheless, when comparing these tests, it is important to remember that they are based on different null hypotheses and thus do not test the same properties.

### 3.3.3 Two-sample Kolmogorov–Smirnov test

The two-sample Kolmogorov-Smirnov (K-S) test is a non-parametric test with the null hypothesis that the samples are drawn from the same distribution. Our local null hypothesis is therefore the following:

$H_{0\,(i,j)}$: $\varphi_{\text{old}(i,j)}$ and $\varphi_{\text{new}(i,j)}$ are drawn from the same distribution.

Here, $\varphi_{\text{old}(i,j)}$ and $\varphi_{\text{new}(i,j)}$ are the samples of the variable $\varphi$ at location $(i,j)$ from the old and new model respectively. The K-S test statistics is given as

$$D_{n_X,n_Y} = \sup_x |F_{X,n_X}(x) - F_{Y,n_Y}(x)|,\tag{7}$$

where $\sup$ is the supremum function and $F_{X,n_X}$ and $F_{Y,n_Y}$ are the empirical distribution functions of the two samples $X$ and $Y$, which is defined as

$$F_{X,n_X}(x) = \frac{1}{n_X}\sum_{i=1}^{n_X} I_{[-\infty,x]}(X_i)\tag{8}$$

with the indicator function $I_{[-\infty,x]}(X_i)$, which is equal to one if $X_i \leq x$ and zero otherwise. The null hypothesis is rejected if

$$D_{n_X,n_Y} > c(\alpha)\sqrt{\frac{n_X + n_Y}{n_X \cdot n_Y}}\,, \tag{9}$$

where $c(\alpha) = \sqrt{-\ln(\frac{\alpha}{2}) \cdot \frac{1}{2}}$ for a given significance level $\alpha$. The K-S test is often perceived to be not as powerful as, for example, the Student's t-test for comparing means and measures of location in general (Wilcox, 1997). However, due to its
different null hypothesis, it might be a more suitable test testing a distribution's shape or spread.

### 3.4   Model description and hardware

The Consortium for Small-scale Modelling (COSMO) model (Baldauf et al., 2011) is a regional model which operates on a grid with rotated latitude-longitude coordinates. It has been originally developed for numerical weather prediction but has been extended to also run in climate mode (Rockel et al., 2008). COSMO uses a split explicit third-order Runge-Kutta discretiza-
tion (Wicker and Skamarock, 2002) in combination with a fifth-order upwind scheme for horizontal advection and an implicit Crank-Nicholson scheme for vertical advection. Parameterizations include a radiation scheme based on the $\delta$-two-stream approach (Ritter and Geleyn, 1992), a single-moment cloud microphysics scheme (Reinhardt and Seifert, 2006), a turbulent kinetic energy based parameterization for the planetary boundary layer (Raschendorfer, 2001), an adapted version of the convection scheme by Tiedtke (1989), a subgrid-scale orography (SSO) scheme by Lott and Miller (1997), and a multi-layer soil
model with a representation of groundwater (Schlemmer et al., 2018). Explicit horizontal diffusion is applied by using a monotonic 4th-order linear scheme acting on model levels for wind, temperature, pressure, specific humidity, and cloud water content (Doms and Baldauf, 2018) with an orographic limiter which helps avoiding excessive vertical mixing around mountains. For the standard experiments in this paper, the explicit diffusion from the monotonic 4th-order linear scheme is set to zero.

Most experiments in this work have been carried out with version 5.09. While COSMO has been originally designed to run on
CPU architectures, this version is also able to run on hybrid GPU-CPU architectures thanks to an implementation described in Fuhrer et al. (2014), which was a joint effort from MeteoSwiss, the ETH-based Center for Climate Systems Modeling (C2SM), and the Swiss National Supercomputing Center (CSCS). The implementation uses the domain-specific language GridTools for the dynamical core and OpenACC compiler directives for the parameterization package. The simulations have been carried out on the Piz Daint supercomputer at CSCS, using Cray XC50 compute nodes consisting of a Intel Xeon E5-2690 v3 CPU and a
NVIDIA Tesla P100 GPU. Except for one ensemble that has been created with a COSMO binary that exclusively uses CPUs, all simulations in this paper have been run in hybrid GPU-CPU mode where the GPUs perform the main load of the work.

### 3.5   Domain and Setup

The domains that have been used for the simulation and verification includes most of Europe and some part of Northern Africa (see Fig. 2). The simulated periods all start on 28 May 2018 at 00:00 UTC and range from several days to 3 months
in length. The initial and the 6-hourly boundary conditions come from the European Centre for Medium-Range Weather Forecasts (ECMWF) ERA-Interim reanalysis (Dee et al., 2011). For this work, we have chosen a $132 \times 129 \times 40$ grid with 50

km horizontal grid spacing and the 40 non-equidistant vertical levels reaching up to a height of 22.7 km. In order to reduce the effect of the lateral boundary conditions, we excluded 15 grid points at each of the lateral boundaries from the verification, resulting in $102 \times 99$ grid points for one vertical layer. As the verification methodology is supposed to be used as a part of an automated testing environment, we have chosen this relatively coarse resolution in order to keep the computational and storage costs low. Running such a simulation for 10 days requires about 4 minutes on one Cray XC50 compute node when using the GPU-accelerated version of COSMO in double-precision. This means that an ensemble of 50 members requires 3 to 4 node hours. However, as the runs can be executed in parallel, the generation of the ensemble is only a matter of minutes.

## 3.6 Experiments

In order to test and demonstrate the methodology, we have performed a series of experiments. Many of these experiments are for cases where we deliberately changed the model. However, we also have one real-world case where we verified the effect of a major update of the supercomputer Piz Daint, on which we have been running our model.

### 3.6.1 Diffusion experiment

COSMO offers the possibility of applying explicit diffusion with a monotonic 4th-order linear scheme with an orographic limiter acting on model levels for wind, temperature, pressure, specific humidity, and cloud water content. Diffusion is applied by introducing an additional operator at the right-hand side of the prognostic equation, similar to

$$\frac{\partial \psi}{\partial t} = S(\psi) + D \cdot c_d \cdot \nabla^4 \psi, \tag{10}$$

where $\psi$ is the prognostic variable, $S$ represents all physical and dynamical source terms for $\psi$, $c_d$ is the default diffusion coefficient in the model, and $D$ is the factor that can be set in order to change the strength of the computational mixing (please refer to Sect. 5.2 in Doms and Baldauf, 2018, for the exact equations including the limiter). By default, we have set $D = 0$, which means that no explicit 4th-order linear diffusion is applied. However, for some experiments we have used $D \in \{0.01, 0.005, 0.001\}$. Such small values should not affect the model results visibly or be easily quantifiable without statistical testing. A value of $D = 1.0$ reduces the amplitude of $2\Delta x$ waves by about a factor $1/3$ per time step. For such a high value, the model results visibly change (Zeman et al., 2021).

### 3.6.2 Architecture: CPU vs GPU

Per default, the simulations shown in this work have been performed with a COSMO binary which makes use of the NVIDIA Tesla P100 GPU on the Cray XC50 nodes (see Sect. 3.4 for details). For this experiment, we have produced an ensemble from the identical source and with identical settings but compiled it to run exclusively on the Intel Xeon E5-2690 v3 CPUs in order to see whether there is a noticeable difference between the CPU version and the GPU version of COSMO.

### 3.6.3 Floating-point precision

In this work, COSMO has been using the double-precision (DP) floating-point format by default, where the representation of a floating-point number requires 64 bits. However, COSMO can also be run in 32 bit single-precision (SP) floating-point representation. The SP version has been developed by MeteoSwiss and is currently used by them for their operational forecasts. They have decided to use the SP version after having carefully evaluated its performance compared to the DP version, which suggests that there are only very small differences. Nevertheless, a reduction of precision leads to greater round-off errors and thus could lead to a noticeable change in model behavior. In order to see whether our methodology would be able to detect differences, we have applied it for a case where the evaluation ensemble has been produced by the SP version of COSMO and the control and reference ensembles by the DP version. It has to be mentioned that for the SP version of COSMO, the soil model and parts of the radiation model are still using double-precision, as some discrepancies were detected during the development of the SP version.

Running COSMO on one node in single-precision, where a floating-point number only requires 32 bits, gives a speedup of around 1.1 for our simulations, most likely due to the increased operational intensity (number of floating-point operations per number of bytes transferred between cache and memory). When running on more than one node, it is often possible to reduce the total number of nodes for the same setup when switching to single-precision, thanks to a drastic reduction of required memory. For example, a model domain and resolution that usually requires four nodes in double-precision (e.g., the same domain as in this paper, but with 12 km grid spacing instead of 50 km grid spacing), often only requires two nodes in single-precision. This results in a coarser domain decomposition and thus fewer overlapping grid cells whose values have to be exchanged between the nodes. Combined with the reduced number of bytes of the floating-point values that have to be exchanged, a significant reduction of data transfer via the interconnect can be achieved, increasing the system's efficiency. While running in SP on only two nodes might be slower than running the same simulation in DP on four nodes, it requires fewer node hours. In this particular case (4 nodes for DP vs. 2 nodes for SP), the speedup in node hours was around 1.4, which makes the use of single-precision an attractive option.

### 3.6.4 Vertical heat diffusion coefficient and soil effects

In order to test the methodology for slow processes related to the hydrological cycle, we have set up an experiment where we induce a relatively small but still notable change. One parameter that has been deemed important to the COSMO model calibration by Bellprat et al. (2016) is the minimal diffusion coefficient for vertical scalar heat transport *tkhmin*. It basically sets a lower bound for the respective coefficient used in the 1D turbulent kinetic energy (TKE) based subgrid-scale turbulence scheme (Doms et al., 2018). By default, we have used a value of *tkhmin* $= 0.35$ for our simulations, but for this evaluation ensemble we have changed it to *tkhmin* $= 0.3$. This is not a huge change, as for example the default value in COSMO is set to *tkhmin* $= 1.0$, whereas the German Weather Service (DWD, Deutscher Wetter Dienst) uses *tkhmin* $= 0.4$ for their operational model with 2.8 km grid spacing (Schättler et al., 2018). The goal of this experiment is to see whether such a change becomes

detectable in the slowly changing soil moisture variable, and if yes, how long it takes to propagate the signal through the different soil layers.

### 3.6.5 No subgrid-scale orography parameterization

So far, the experiments have been set up for cases where there are only slight model changes. In order to see whether the methodology is able to reject results from significantly different models confidently, we have applied it on an evaluation ensemble where the model had the subgrid-scale orography (SSO) parameterization by Lott and Miller (1997) switched off. At a grid-spacing of 50 km, orography cannot be realistically represented in a model, which is why the parameterization should be switched on in order to account for orographic form drag and gravity wave drag effects. Zadra et al. (2003) and Sandu

et al. (2013) both showed improvements in both short- and medium-range forecasts with a SSO parameterization based on the formulation by Lott and Miller (1997) for the Canadian Global Environmental Mutiscale (GEM) model and the ECMWF Integrated Forecast System (IFS). Pithan et al. (2015) showed that the parameterization was able to significantly reduce biases in large-scale pressure gradients and zonal wind speeds in climate runs with the general circulation model ECHAM6. So we expect the test to clearly reject the global null hypothesis within the first few days, but also for a longer period of time, which

is why we use model runs of 90 days for this experiment.

### 3.6.6 Piz Daint Update

The supercomputer Piz Daint at the Swiss National Supercomputing Center (CSCS) has recently received two major updates on 9 September 2020 and 16 March 2021. The major changes that affected COSMO were new versions of the Cray Programming Toolkit (CDT), which changed the compilation environment for COSMO, with the new version being CDT 20.08 compared to

430 the old version CDT 19.08 before the first update in September 2020. Both changes were associated with the loss of bit-identical execution. Using containers, CSCS created a testing environment that replicated the environment before the first update on 9 September 2020 with CDT 19.08. With this environment, we could reproduce the results from runs before the update in a bit-identical way. So by using this containerized version and comparing its output to the output from the executable that has been compiled in the updated environment with CDT 20.08, we were able to apply our methodology for a realistic scenario

with typical changes in a model development context. Indeed, the system upgrade of the Piz Daint software environment was the motivation for the current study.

## 4 Results

### 4.1 Diffusion experiment

Here, we discuss the results from the diffusion experiment described in Sect. 3.6.1. Figure 2 shows why it is important to have

440 such a statistical approach for verification. By just looking at the mean values of the ensembles and their differences (in this case, 850 hPa temperature), it is impossible to say whether the two ensembles come from the same distribution. There are some

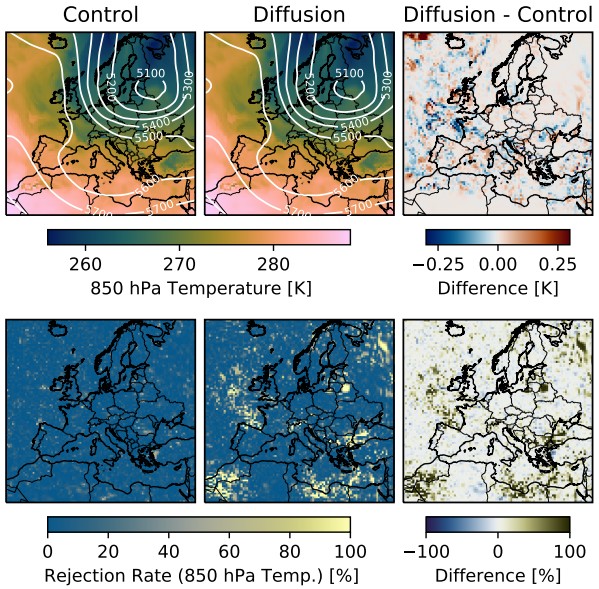

**Figure 2.** The top row shows the ensemble-mean 850 hPa temperature (color shading) and 500 hPa geopotential height (white contours) for the control (left) and diffusion ensemble with $D = 0.01$ (middle) after 24 hours, using $n_E = 50$ members per ensemble. The difference in mean temperature is shown in the top right panel. The bottom row shows the mean rejection rate for 850 hPa temperature (calculated with the MWU test for $m = 100$ subsamples with $n_S = 20$ members per subsample) for each grid cell for these two ensembles, as well as their difference. The substantial differences in the mean rejection rates indicate clearly that the two ensembles come from different models.

small differences, but these could also be a product of internal variability, and the tiny amount of additional explicit diffusion in the diffusion ensemble ($D = 0.01$) is not visible by eye. However, the mean rejection rates calculated with the methodology are clearly higher for the diffusion ensemble in some places in comparison to the control, indicating that the ensembles do

not come from the same model. This becomes clear when we compare the mean rejection rate for 500 hPa geopotential of the diffusion ensemble with $D = 0.005$ to the 0.95 quantile of the control at the bottom of Fig. 3. The methodology can reject the global null hypothesis for the first 60 hours. Afterward, it is no longer able to reject it, which indicates that from this point on, the effect of internal variability is greater than the one from the additional explicit diffusion.

In Fig. 3 (top panel) , we can also see that the mean rejection rate of the control is very close to the expected 5%, which is the

450 significance level $\alpha$ of the underlying MWU test. However, the rejection rate of some samples in the control deviate by quite a bit from 5% even though the results come from an identical model. Generally, the spread of the rejection rates also becomes bigger with time, which likely is related to changes in spatial correlation and/or decreasing predictability. While the initial perturbations are random and therefore not spatially correlated, the statistical independence becomes already invalid after the first time step, as a perturbation of a value in a grid cell will naturally affect the corresponding values in the neighboring

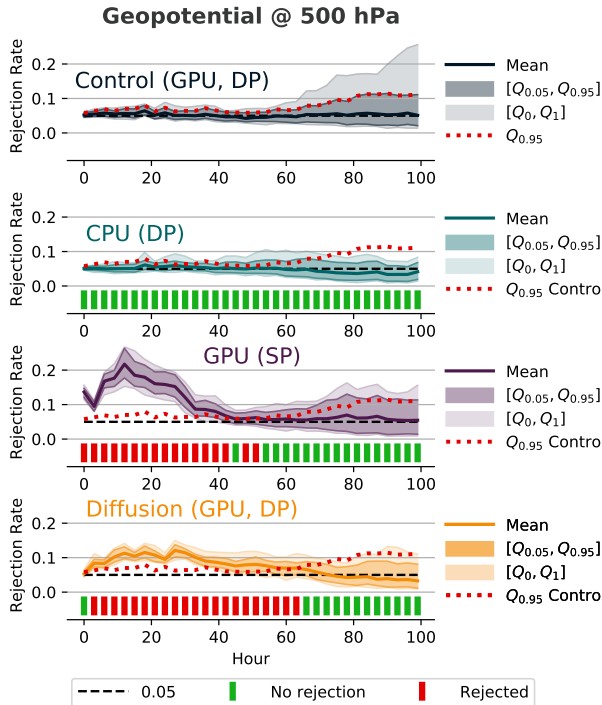

**Figure 3.** Rejection rates and decisions for $H_{0\,(\text{global})}$ for $500\,\text{hPa}$ geopotential using the MWU test as a underlying statistical hypothesis test with an ensemble size of $n_E = 100$ and $m = 100$ randomly drawn subsamples with a subsample size of $n_S = 50$. The reference and control ensemble were produced by COSMO running on GPUs in double-precision (top), and it was compared against (from top to bottom) COSMO running on CPUs in double-precision, on GPUs in single-precision, and on GPUs in double-precision with additional explicit diffusion ($D = 0.005$). We reject the null hypothesis if the mean rejection rate is above the 95th percentile of the rejection rate distribution from the control ensemble (red dotted line). The test detects no differences for the CPU version in DP, but it detects differences for the other two ensembles during the first few hours/days. The rejection for the initial conditions of the SP ensemble is most likely associated with differences in the diagnostic calculation of the geopotential due to the reduced precision.

grid cells. This increasing spread emphasizes the importance of having such a control rejection rate for the decision on the evaluation ensemble.

The first two columns of Fig. 4 show the global decisions for 16 output fields for the diffusion experiment with $D = 0.005$ and $D = 0.001$. We believe that such a set of variables offers a good representation of the most important processes in an atmospheric model (i.e., dynamics, radiation, microphysics, surface fluxes) and, considering the often high correlation between different variables, is therefore likely sufficient to detect all but the tiniest changes in a model. While all variables seem to be affected for the ensemble with a larger diffusion coefficient, the smaller diffusion coefficient leads to a smaller but still noticeable number of rejections for many variables.

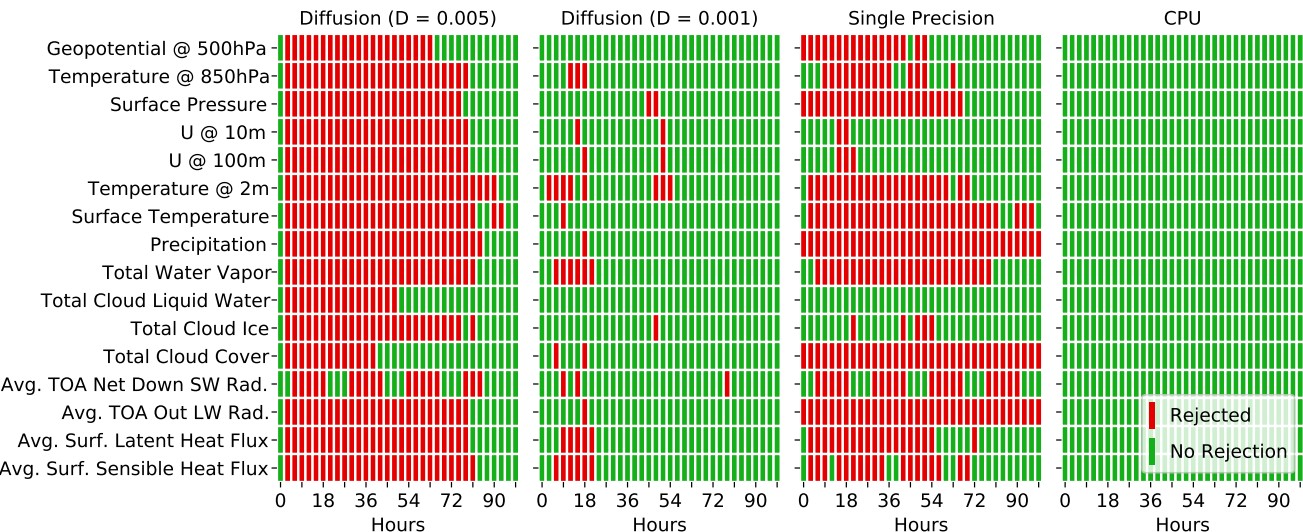

**Figure 4.** Global decisions for several variables for two ensembles with additional explicit diffusion with $D = 0.005$ and $D = 0.001$, the single-precision ensemble, and an ensemble from the CPU version of COSMO using the MWU test with $n_E = 100$, $n_S = 50$, and $m = 100$. A smaller diffusion coefficient clearly leads to fewer rejections. The CPU ensemble shows no rejections of the tested variables, meaning that the GPU and CPU executables cannot be distinguished.

## 4.2 Architecture: CPU vs GPU

The COSMO executable running on CPUs does not lead to any global rejections compared to the executable running mainly on GPUs, which is exemplified in Fig. 3 for 500 hPa geopotential and for all 16 tested variables in the fourth column in Fig. 4. So while the results are not bit-identical, we consider the difference between these two executables negligible. This confirms that the GPU implementation of the COSMO model is of very high quality, as in terms of execution, it cannot be distinguished from the original CPU implementation. This bespeaks an impressive achievement given that the whole code (dynamical core and parameterization package) had to be refactored.

## 4.3 Floating-point precision

The results of the verification of the single-precision (SP) version of COSMO against the corresponding double-precision (DP) version can be seen in Fig. 3 for the 500 hPa geopotential and in Fig. 4 for all 16 tested variables. Before discussing the results, we remind the reader that some of the variables, notably in the soil model and the radiation codes, are retained in double-precision, as some discrepancies were detected during the development of the SP version. When looking at Fig. 3 (third panel), it should be noted that the geopotential is a plain diagnostic field in the COSMO model, so it is not perturbed initially but diagnosed at output time from the prognostic variables. However, as the geopotential is vertically integrated, it encompasses information from many levels and variables and can thus be considered a well-suited field for testing. One of the most striking

features in Fig. 3 is that the methodology rejects the SP version already at the initial state of the models. At this state, the perturbation has already been applied according to Eq. (3), but the model has performed only one time step. This one time step before the initial output has to be performed in COSMO to compute the diagnostic quantities. Typically, one time step is not enough time for small differences to manifest themselves, as can be seen by the lack of rejections at hour zero for the diffusion ensemble in Fig. 3 and Fig. 4. It is not entirely clear why the 500 hPa geopotential rejection rate is that high after one time step for the SP ensemble, but we assume a small difference in its calculation due to increased round-off errors for the vertical integration. Considering that the small perturbations did not have much time to grow, there is no real internal variability that could "hide" that difference. After 3 hours, the mean rejection rate of the SP ensemble is substantially lower but still higher than the 0.95 quantile from the control. Afterward, the rejection rate increases again and follows a similar trajectory as the diffusion ensemble's but with a higher magnitude. In order to rule out differences in perturbation strength due to rounding errors (see also Sect. 3.2), we have performed the same experiment for a modified double-precision version of COSMO, where the to-be perturbed fields are cast to single-precision, the perturbation is applied in single-precision, and the fields are then cast back to double-precision. However, this had no effect on the results, and the SP ensemble was still rejected with the same magnitude for the initial conditions.

The third column of Fig. 4 shows the global decisions for 16 output variables of the single-precision ensemble during the first 100 hours. Overall, the number of rejections is similar to the one of the diffusion ensemble with $D = 0.005$ (first column). However, while most variables show a similar rejection pattern for the diffusion ensemble, the switch to single-precision does not affect all variables to the same extent. Next to 500 hPa geopotential, the test also rejects other variables after only one time step. The rejections of the diagnostic surface pressure, total cloud cover, and average top-of-atmosphere (TOA) outgoing longwave radiation are probably also caused by differences in the diagnostic calculations due to the reduced precision. The precipitation variable represents the sum of precipitation during the last hour. After the first time step, the model has only produced very little precipitation. In this case, the maximum precipitation amount per grid point is below $0.09\,\mathrm{mm\,h^{-1}}$ in all ensemble members of the DP and SP ensemble. Therefore, it is possible that the increased round-off error by the single-precision representation of very small numbers may lead to the rejection for precipitation at hour zero.

## 4.4 Statistical hypothesis tests

We have tested our methodology with the different statistical hypothesis tests described in Sect. 3.3 for the test case with additional explicit diffusion (see above). Figure 5 shows the respective rejection rates and decisions for several variables. The rejection rates from the Student's t-test and the MWU test are almost identical for all variables shown here. This confirms the robust behavior of the Student's t-test, despite violations of the normality assumptions. The results especially exemplify this for precipitation, where the means of the distribution do not follow a normal distribution and are floored (no negative precipitation). Like the MWU test, the K-S test is non-parametric and therefore does not rely on assumptions about the distribution of the variables. However, its rejection rate is generally lower than that of the MWU test and the Student's t-test. This effect can also be seen in the 0.95 quantile of the control rejection rate, which is generally lower than for the other two hypothesis tests. The lower rejection rate is most likely associated with the lower power of the K-S test (see Sect. 3.3.3). However, the decision

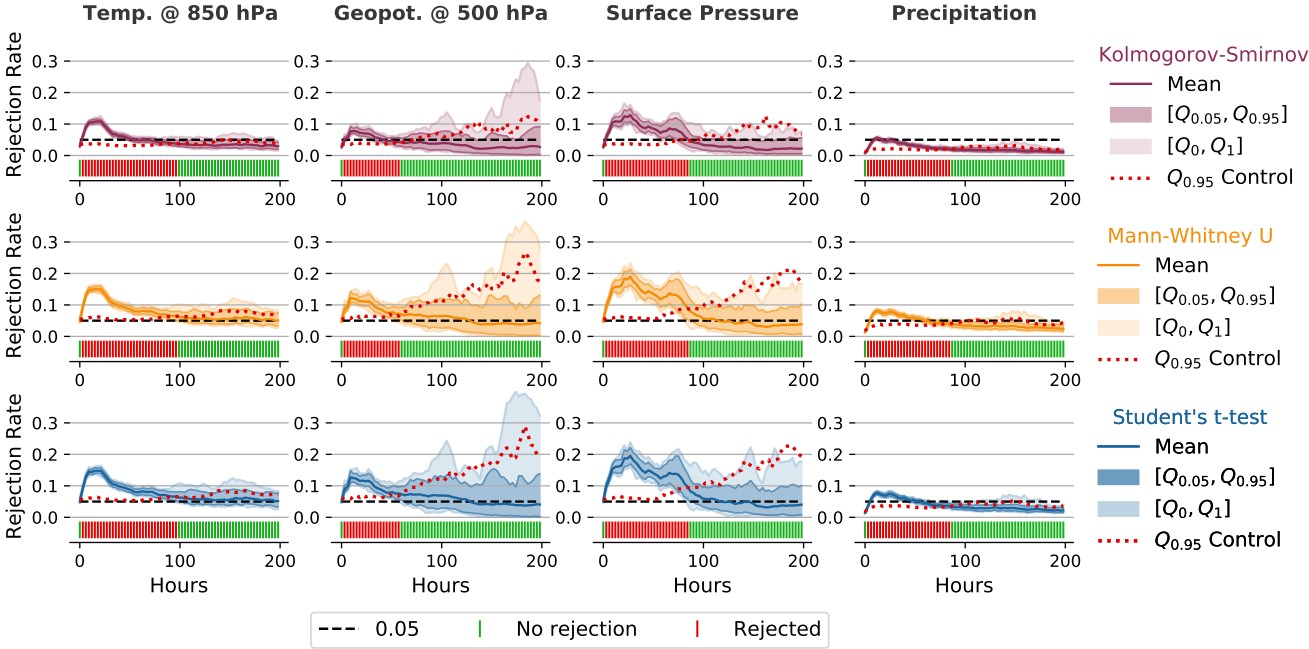

**Figure 5.** Rejection rates and decisions similar to Fig. 3 for different variables and with the use of different underlying statistical hypothesis tests for the diffusion ensemble with $D = 0.01$ and $n_E = 50$, $n_S = 20$, and $m = 100$. While the rejection rates show some differences, the global decisions are very similar throughout all tests for the corresponding variables. The rejection rates with the K-S test are usually lower than for the other two tests, but this does not affect the global decisions, as the respective 0.95 quantiles from the control ensemble are also lower. The Student's t-test shows very similar rejection rates as the non-parametric MWU test, even for precipitation, which is clearly not normally distributed.

(reject or not reject) is always the same in this case for all tests. This indicates that any of these tests is suitable as an underlying statistical hypothesis test and that the choice of the statistical test is not very critical for our methodology. Nevertheless, we have decided to use the MWU test for most of the subsequent experiments, as it offers a slightly higher rejection rate than the K-S test and, as a non-parametric test, its use is easier to justify than the use of the Student's t-test, even though these two produce almost identical results.

### 4.5 Vertical heat diffusion and soil effects

Figure 6 shows the rejections rates and global decisions for the 2 m temperature and soil moisture at different depths for the model setting with a modified minimal diffusion coefficient for vertical scalar heat transport (*tkhmin* = 0.3 instead of 0.35). Note that this change will only affect a subset of the grid points, as *tkhmin* represents a limiter. The rejection rate is quite high for the 2 m temperature during the first few days. For the soil moisture at different depths, we can see that the magnitude of the rejection rate decreases the deeper we go. Furthermore, the initial perturbation and the subsequent internal variability

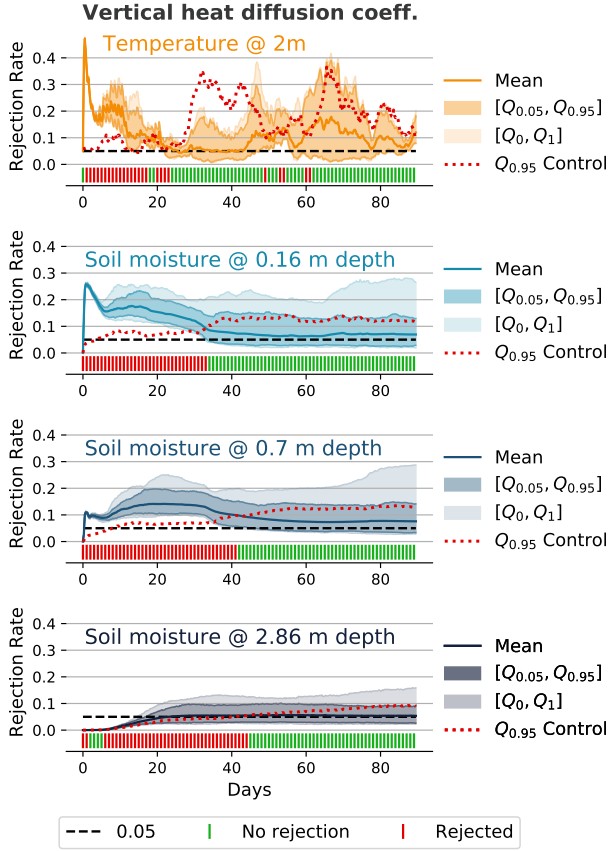

**Figure 6.** Rejection rates and decisions similar to Fig. 3 for 2 m temperature and soil moisture at different depths for an ensemble where the minimal diffusion coefficient for vertical scalar heat transport has been slightly changed (*tkhmin* = 0.3 instead of 0.35) with $n_E = 50$, $n_S = 20$, and $m = 100$. The initial random perturbation of the atmosphere needs some time to travel to the deeper soil layers. While the magnitude of the rejection rate is significantly lower for the deeper soil layers, the difference is noticeable for a longer period of time.

of the atmosphere need some time to travel to the lower layers, which is most obvious in the layer at 2.86 m depth. In this layer, the rejection rate remains close to zero for the first few days because there is almost no difference visible between the different ensemble members. As a consequence of the not yet "arrived" perturbation, the global decision for this layer should be interpreted with caution during these first few days. However, while the magnitude and the variability of the rejection rate decrease for the lower soil layers, the effect is visible for longer, which is most probably related to the slower processes in the soil. For 2 m temperature, there are still some rejections after 50-60 days. However, the test is usually not able to reject the global null hypothesis for 2 m temperature after 25 days, which indicates that from this point on, the effect from the change of *tkhmin* has been overshadowed by internal variability or that the test might no longer be sensitive enough to detect the difference with such a small ensemble and subsample size ($n_E = 50$, $n_S = 20$, $m = 100$).

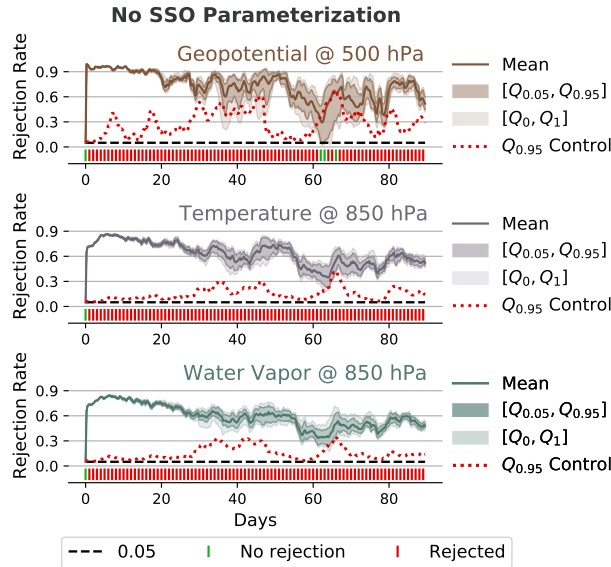

**Figure 7.** Rejection rates and decisions similar to Fig. 3 but for the 500 hPa geopotential, 850 hPa temperature, and 850 hPa water vapor amount and for an evaluation ensemble where the subgrid-scale orography (SSO) parameterization has been switched off ($n_E = 50$, $n_S = 20$, $m = 100$). The methodology rejects the null hypothesis throughout all 90 days, except in three instances for the 500 hPa geopotential. The difference between the mean rejection rate of the evaluation ensemble and the 0.95 quantile of the control is quite large and persistent (also considering the relatively small ensemble and subsample sizes), which indicates that such a big change in the model is detectable for an even longer time.

## 4.6 No subgrid-scale orography parameterization

Disabling the SSO parameterization is a substantial change, and our methodology can detect this for the whole three months simulation time. Despite the relatively small ensemble size of $n_E = 50$ and subsample size of $n_S = 20$, the mean rejection rate for the three variables shown in Fig. 7 is very high and seems to remain at a relatively constant level after the first month. This indicates that the difference would also be detectable after a longer simulation time, even though the variability on a grid cell level must be very high.

## 4.7 Piz Daint Update

Figure 8 shows that we do not detect any differences after the update of the supercomputer Piz Daint. This test was one of the first cases where the methodology has been used and it was performed with a relatively low number of ensemble and subsample members ($n_E = 50$, $n_S = 20$). However, considering how closely the 0.95 quantile from the control ensemble follows the 0.95 quantile from the evaluation ensemble and how close the mean rejection rate from the evaluation ensemble is to 0.05, we believe that also a test with a higher number of ensemble and subsample members would either show no rejections or, for much

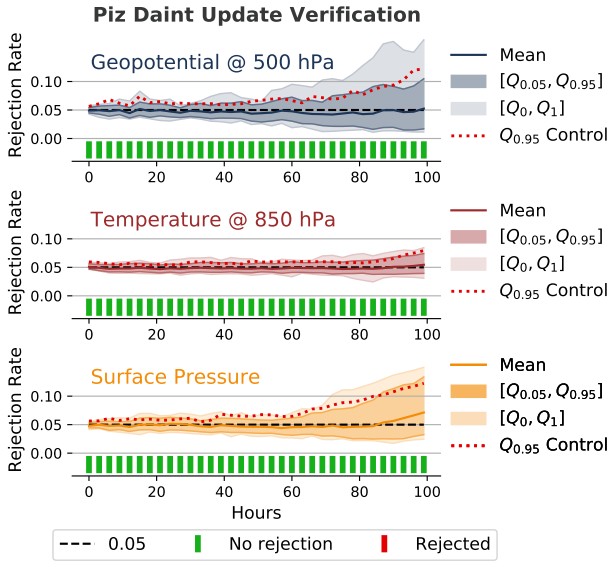

**Figure 8.** Rejection rates and decisions similar to Fig. 3 for the 500 hPa geopotential, 850 hPa temperature, and surface pressure from the verification of a major system update of the underlying supercomputer Piz Daint. The methodology cannot reject the null hypothesis (at least not for the used ensemble size of $n_E = 50$, subsample size of $n_S = 20$, and $m = 100$ subsamples), which suggests that the update did not significantly affect the model behavior.

larger ensemble and subsample sizes, a number of rejections that is comparable to the expected number of false positives (see
Sect. 4.10).

### 4.8 Sensitivity to ensemble and subsample sizes

In order to test the sensitivity of the methodology to the number of ensemble members $n_E$, the number of subsample members $n_S$, and the number of subsamples $m$, we have performed the test for the diffusion experiment with $D = 0.005$ for a combination of different values for $n_E$, $n_S$, and $m$. Figure 9 shows the effect of different ensemble and subsample sizes on the evaluation of 500 hPa geopotential. More ensemble and subsample members increase the test's sensitivity, whereas a higher number of subsamples ($m = 500$ instead of $m = 100$) has a negligible effect (not shown in the figure), which indicates that using 100 subsamples is sufficient for this methodology.

### 4.9 Influence of spatial averaging

Most existing verification methodologies for weather and climate models involve some form of spatial averaging of output variables (see Sect. 2.1). Our methodology evaluates the atmospheric fields at every grid point on a given vertical level. The idea behind this more fine-grained approach is that it should allow us to identify differences in small-scale features that may

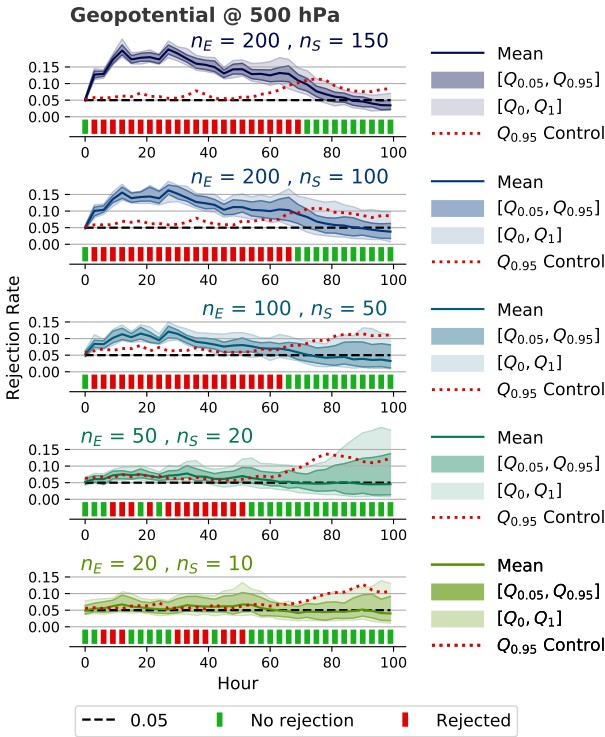

**Figure 9.** Rejection rates and decisions for 500 hPa geopotential as in Fig. 3 for the diffusion ensemble ($D = 0.005$) with different numbers of ensemble members $n_E$, subsample members $n_S$, and $m = 100$ subsamples. Larger values for $n_E$ and $n_S$ increase the sensitivity of the methodology.

not affect spatial averages. In order to evaluate this, the model output from some of the previous experiments has been spatially averaged into tiles consisting of an increasing number of grid cells ($1 \times 1$, $2 \times 2$, $4 \times 4$, $8 \times 8$, and $16 \times 16$ grid cells per tile).

Figure 10 shows the rejection rates for two diffusion ensembles ($D = 0.005$ and $D = 0.001$), the CPU ensemble, and an
ensemble that was obtained from an identical model the same way as the control ensemble. The rates represent the fraction of global rejections from the 16 variables during the first 100 hours (i.e., the fraction that is red in Fig. 4), and they have been calculated for different tile sizes and numbers of ensemble and subsample members. For the diffusion ensembles, the spatial averaging reduces the test's sensitivity for all ensemble and subsample sizes. These results strongly indicate that a test on a grid cell level might detect differences that would not be detected by methods that compare domain mean values or use some
other form of spatial averaging.

For the CPU ensemble, we only see a rejection rate that is significantly higher than zero for the largest subsample size in Fig. 10. However, since the rejection rate is similar to the corresponding false positive rate, one cannot reject the null hypothesis. It is also interesting to see that spatial averaging does not affect the rejection rate of the CPU ensemble and the ensemble that has been used to calculate the number of false positives.

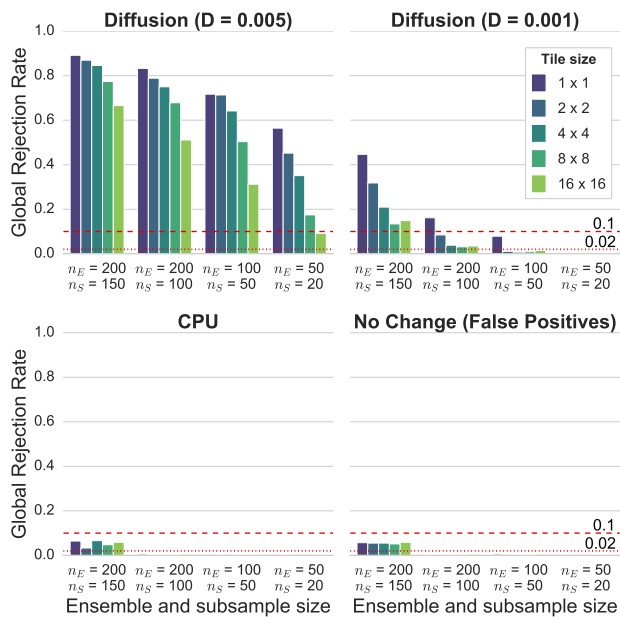

**Figure 10.** Global rejection rates of the 16 variables during the first 100 hours, as in Fig. 4, for the diffusion ensemble with $D = 0.005$. A rate of 1.0 would mean that all global decisions would show a rejection (i.e., only red in Fig. 4). The rates have been calculated for different ensemble and subsample sizes with $m = 100$ randomly drawn subsamples. They are grouped by tile size, where one tile represents the spatial average value of $n \times n$ grid cells. Spatial averaging clearly reduces the sensitivity of the test for all ensemble sizes. The red lines indicate thresholds that could be used for an automated testing framework. For example, based on the false positive rate for $n_E = 200$ and $n_S = 150$, one could define a rejection rate of 0.1 as a threshold for this combination of ensemble and subsample size (i.e., the model has significantly changed if the rejection rate is greater than 0.1). The threshold should be lower for smaller ensemble and subsample sizes (e.g., 0.02 for $n_E = 200$ and $n_S = 100$).

## 4.10 False positives and determining a threshold for automated testing

Looking at the rejection rates of the ensemble with no change in Fig. 10 (bottom right), we can see that we have almost no false positives except for $n_E = 200$ and $n_S = 150$. The reason for this is likely a combination of a lower variability of the result for larger subsample sizes (i.e., the test becomes more accurate) as well as the fact that with $n_S = 150$ and $n_E = 200$ many subsamples will consist of a set of very similar ensemble members, which also reduces the variability of the result. This effect can also be seen in Fig. 9, where the 0.95 quantile is quite close to the mean rejection rate for $n_E = 200$ and $n_S = 150$. This "narrow" distribution of rejection rates likely increases the probability of the mean rejection rate of the false positive ensemble being higher than the 0.95 quantile of the rejection rate of the control ensemble.

While the false positive rate for the smaller ensemble and subsample sizes is very close to zero with our methodology, we still have to expect a certain amount of false positives. An automated testing framework requires a clear pass/fail decision, and ideally, the test should not fail because of false positives. The false positive rate depends on the ensemble and subsample

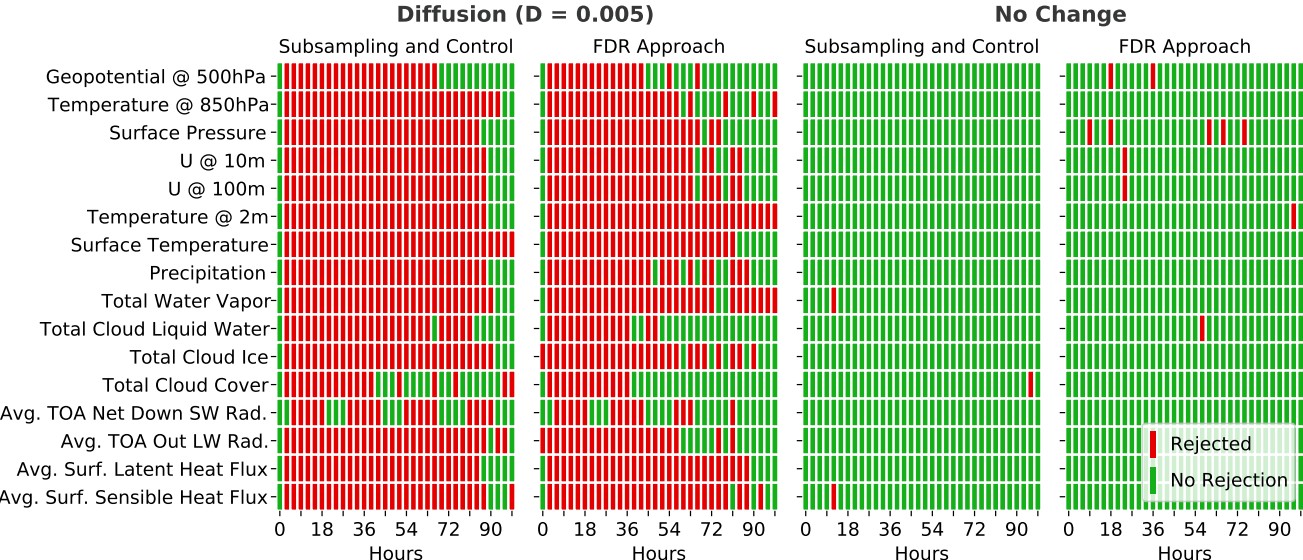

**Figure 11.** Comparison between our methodology, using subsampling ($n_E = 100$, $n_S = 50$, $m = 100$) and a control ensemble, and an approach that uses only one comparison between all members of the two ensembles with $n_E = 100$ in combination with the FDR correction and $\alpha_{\text{FDR}} = 0.05$. The Student's t-test has been used for the local hypothesis testing in both cases. The first two columns show the global rejections for the diffusion ensemble ($D = 0.005$), whereas the last two columns show the respective rejections for an ensemble from an identical model (no change) to compare the false positive rate. Both methods show similar rejections with a slightly higher number of false positives for the FDR approach.

size, the evaluated variables, and the evaluation period. In order to determine a reasonable rejection rate threshold for the given parameters, the test should be first performed on an ensemble from a model that is identical to the reference and the control ensemble. Based on the results in Fig. 10 for the output without spatial averaging, we would for example set the threshold to 0.1 (dashed red line) for $n_E = 200$ and $n_S = 150$. For $n_E = 200$ and $n_S = 100$, a threshold of 0.02 would probably make sense (dotted red line), and we could go even lower for smaller ensemble and subsample sizes.

### 4.11 Comparison with the FDR method

The approach in our methodology, which is based on subsampling and a control ensemble, is an effective way to determine field significance while accounting for spatial correlation and reducing the effect of false positives. As already discussed in Sect. 2.2, the FDR approach by Benjamini and Hochberg (1995) serves a similar purpose by limiting the fraction of false rejections out of all rejections. The big advantage of the FDR approach is that we only need two ensembles (no control ensemble) and no subsampling, which reduces the computational costs. Figure 11 shows the global rejections of our methodology (with $n_E = 100$, $n_S = 50$, and $m = 100$) and the FDR approach that only performs one comparison with $n_E = 100$ (no subsampling) and $\alpha_{\text{FDR}} = 0.05$. We use the Student's t-test as a local null hypothesis test for both methods. The FDR approach shows a similar

result for the diffusion ensemble with $D = 0.005$ as our approach with a control ensemble and subsampling. With the FDR approach, the number of false positives is larger by a factor of 3 to 4, but one could account for this by using a slightly higher threshold for the global rejection rate (see previous section). This would slightly reduce the test's sensitivity, but considering the FDR approach's lower computational cost, it seems to be an attractive alternative to our approach, especially for frequent automated testing.

## 5  Discussion

As opposed to most existing verification methodologies described in Sect. 2, our methodology does not rely on any averaging in either space or time. This approach offers several advantages. The verification on a grid-cell level allows us to identify differences in small-scale and short-lived features that may not affect spatial or temporal averages. Furthermore, it provides fine-grained information in space and time and therefore gives helpful information for investigating the source of the difference. A good example of this is the initial rejection of some diagnostic fields, such as 500 hPa geopotential, for the single-precision experiment. The test rejects the null hypothesis already after one single time step, which indicates that there are already detectable differences in the diagnostic calculation of the respective field (see Sect. 4.3 for further detail). The focus on instantaneous values or averages over a small time frame is also a way to consider internal variability. Minor differences can often only be detected during the first few hours or days before the increasing internal variability outweighs the effect of the change. Therefore, we think short simulations of a few days should generally be preferred to longer, computationally more expensive simulations.

It is not entirely clear how sensitive such a methodology is in detecting differences in long climate simulations. For the verification of very slow processes, longer simulations with either spatial or temporal averaging might appear to be the better choice. However, the current methodology using short integrations can also detect changes in slower variables such as soil moisture within the first few days, which indicates that it might also be suited for climate simulations. Moreover, given that differences from the frequent changes (e.g., compiler upgrades, library updates, and minor code rearrangements) typically manifest themselves already early in the simulation (see Milroy et al., 2018), we think that this is a reasonable approach with low computational costs. Nevertheless, it is worthwhile to rethink our methodology in the case of a global coupled climate model that may represent very fast (e.g., the atmospheric model) and very slow (e.g., an ice sheet model) components. In such a case, it might be advantageous to test the different model components in stand-alone mode, possibly using different integration periods, before evaluating the fully coupled system focusing on the variables heavily affected by the coupling (e.g., near-surface temperature for ocean-atmosphere coupling). However, further studies on this topic would be needed.

The methodology clearly shows some sensitivity with regard to the ensemble and subsample size. A larger number of ensemble and subsample members generally increases the test's sensitivity but will also lead to higher computational costs. Similarly, the choice of the tested variables also has to be considered. Testing all possible model variables at all vertical levels would guarantee the highest degree of reliability. However, this is unfeasible due to the high computational costs it would demand. Moreover, since the atmosphere is such a complex and interconnected system, many variables are highly correlated.

Therefore, and based on our results, we think that testing a few standard output variables on selected vertical levels (as in Fig. 4) is already sufficient for all but the tiniest changes.

# 6 Conclusions and outlook

We presented an ensemble-based verification methodology based on statistical hypothesis testing to detect model changes objectively. The methodology operates on a grid-cell level and works for instantaneous and accumulated/averaged variables. We showed that spatial averaging lowers the chance to detect small-scale changes such as diffusion. Furthermore, the study suggests that short-term ensemble simulations (days to months) are best suited, as the smallest changes are often only detectable during the first few hours of the simulation. Combined with the fact that the methodology already works well for coarse

resolutions (here 50 km grid spacing), the methodology is a good candidate for a relatively inexpensive automated system test. We showed that the choice of the underlying statistical hypothesis test is secondary, as long as the rejection rate is compared to a rejection rate distribution from a control ensemble that has been generated with an identical statistical hypothesis test.

While the methodology could theoretically be applied to all model output variables at all vertical levels and thus be exhaustive, we think that this would be overkill. Based on our results using a limited-area climate model and the high correlations

between many atmospheric variables, we think that a set of key variables that reflect the most important processes in an atmospheric model might already be sufficient to cover most of the atmospheric and land-surface processes. However, for a fully-coupled global climate model, further considerations will be needed.

The verification methodology detected several configuration changes, ranging from very small changes, such as tiny increases in horizontal diffusion or changes in the minimum vertical heat diffusion coefficient, to more substantial changes, such

as disabling the subgrid-scale orography (SSO) parameterization. The test was not able to detect any differences between the regional weather and climate model COSMO running on GPUs or on CPUs on the same supercomputer (Piz Daint, CSCS, Switzerland). However, the test detected differences between single- and double-precision versions of the model for almost all tested variables. In the case of single versus double precision analysis, rejections occur already after one single time step for some diagnostic variables, suggesting precision-sensitive operations in the diagnostic calculation. Furthermore, the methodol-

ogy has already been successfully applied for the verification of the regional weather and climate model COSMO after a major system update of the underlying supercomputer (Piz Daint, CSCS, Switzerland).

Nonetheless, the results of such a test have to be interpreted with caution and might give a false sense of security. On the one hand, there are potential issues with any statistical hypothesis test, as the inability to reject the null hypothesis does not automatically mean that it is true. On the other hand, even though verification is termed a "system test", it is hardly

possible to test the whole model. There are countless configurations for such models, and testing all these configurations (i.e., different physical parameterizations, resolutions, numerical methods) is almost impossible and would require a substantial computational effort. The methodology also has some potential limitations if a certain part of the code is only very rarely activated (as potentially with threshold-triggered processes). First results also show that the FDR approach seems to be a suitable and computationally less expensive alternative to using a control ensemble and subsampling to determine the field

significance of spatially correlated output data. However, the FDR approach has a somewhat higher rate of false rejections, and thus a somewhat lower sensitivity.

For future work, we intend to apply the methodology for more test cases, such as the compilation of the model with different optimization levels or running the model on different supercomputers. It would also be interesting to directly compare our verification methodology to other already existing methodologies to understand better the differences in sensitivity and applicability.

*Code and data availability.* The source code that has been used to calculate the rejection rates shown in this paper is available under https://doi.org/10.5281/zenodo.6355694. The corresponding model output data from the shorter ensemble simulations (5 days) is available under https://doi.org/10.5281/zenodo.6354200 and https://doi.org/10.5281/zenodo.6355647. The COSMO model that has been used in this study is available under license (see http://www.cosmo-model.org/content/consortium/licencing.htm for more information, last access: 15 January 2022). COSMO may be used for operational and for research applications by the members of the COSMO consortium. Moreover, within a license agreement, the COSMO model may be used for operational and research applications by other national (hydro-)meteorological services, universities, and research institutes. ERA-Interim reanalysis data, which has been used for initial and lateral boundary conditions, is available at https://www.ecmwf.int/en/forecasts/datasets/reanalysis-datasets/era-interim (last access: 15 January 2022).

## Appendix A: Influence of perturbation strength

As described in Sect. 3.2, we have chosen a relatively strong initial perturbation for ensemble generation with a magnitude in the order of $10^{-4}$. Most other existing verification frameworks use a weaker perturbation with a magnitude in the order of $10^{-14}$ (e.g. Baker et al., 2015; Mahajan et al., 2017; Milroy et al., 2018). For us, the chosen perturbation magnitude proved to be a good compromise between not disturbing the initial conditions too much but still providing a good enough ensemble spread for the statistical verification during the first few hours. Furthermore, choosing such a relatively strong perturbation also allows us to examine the effects of single versus double-precision floating-point representation, as the choice minimizes the chance of undesirable rounding artifacts already for the perturbation.

Figure A1 shows that the mean coefficient of variation averaged over all grid points of 850 hPa temperature, which is one of the directly perturbed variables, is not substantially higher with $\epsilon = 10^{-4}$ than with $\epsilon = 10^{-16}$ during the first few days. After around 300 hours, the influence of the perturbation strength seems to be negligible.

*Author contributions.* CZ and CS conceptualized the verification methodology and designed the study. CZ performed the COSMO model ensemble simulations and developed the code for the verification of the model results. CZ wrote the paper with contributions from CS.

*Competing interests.* The authors declare that they have no conflict of interest.

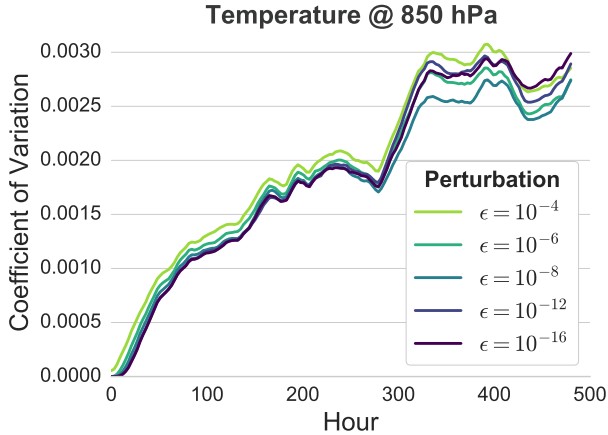

**Figure A1.** Mean coefficient of variation averaged over all grid points of 850 hPa temperature from ensembles (50 members per ensemble) with different initial perturbation magnitudes according to Eq. (3). The relatively strong perturbation used in this work ($\epsilon = 10^{-4}$) only leads to a slightly higher variance during the first few days than a perturbation at machine precision ($\epsilon = 10^{-16}$).

*Acknowledgements.* We would like to thank the two anonymous reviewers for their valuable comments. We acknowledge PRACE for awarding compute resources for the COSMO simulations on Piz Daint at the Swiss National Supercomputing Centre (CSCS). We also acknowledge the Federal Office for Meteorology and Climatology MeteoSwiss, CSCS, and ETH Zurich for their contributions to the development of the GPU-accelerated version of COSMO. In the discussion leading to this paper, we benefited from useful comments of several ETH, MeteoSwiss, and CSCS colleagues.

690

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
