# Peer review of "An Ensemble-Based Statistical Methodology to Detect Differences in Weather and Climate Model Executables"

_Geoscientific Model Development, 2021_

## Author Comment (AC1)

**Answers to Reviewer Comments 1:**

Many thanks for reviewing our paper and the constructive comments on how we could improve it. Please find our answers to the specific comments below.

The authors describe an ensemble verification method that uses statistical hypothesis testing to assess the effects of minor model changes. Note that other statistical ensembles tests have been previously proposed (which the authors reference) and are in use for evaluating climate models. These types of statistical evaluations are important as bit-reproducibility is not a practical approach for verification (or correctness checking) given the chaotic nature of climate/weather models as well as modern heterogeneous architectures.

The authors' statistical ensemble approach uses a total of three ensembles: the so-called "control" and "reference" ensembles are both from the old or accepted model, and an "evaluation" ensemble is generated from the new model or configuration that is being evaluated for differences. It is important to note that their statistical tests are applied locally -- meaning at each grid cell.  The difficulty here is that the local statistical tests can't be assumed to be independent because the data is very likely spatially correlated. (And climate models have many different variables, many of which will have different spatial correlation properties). Therefore, how does one decide on the global reject criteria? The authors rightly admit that this would be non-trivial to determine for all the variables, and for this reason, they generate the two ensembles from the same (old) model (control and reference) to compare and empirically determine the rejection rate distribution. This rejection rate distribution computed from the two ensembles (which are not different) then becomes the baseline against which the rejection rate from the new (evaluation) ensemble as compared to the reference ensemble can be evaluated (nicely illustrated in Figure 1 in the paper). On the whole, this process seems rather involved, though the authors do a nice job explaining their approach.   Results are given for a number of experiments for which the anticipated output is then confirmed by the approach. Unfortunately, the authors do not compare their test to other available ensemble statistical tests. Of course a comparison would be useful as well as interesting (the authors acknowledge this), but perhaps the amount of additional effort required is too high...

We do agree that a comparison with already existing statistical tests using ensemble simulations would be very interesting and useful. However, this would require a significant additional effort and we therefore believe it to be beyond the scope of this work. We think that such an evaluation could for example be part of a greater intercomparison project, where the existing methodologies are compared for different cases.

One aspect of this work that I do find concerning is that I do not have a sense of the robustness of this approach. In particular, with ensemble methods, the size of the ensemble typically directly influences the robustness of the method. In this case, I would expect that a much larger ensemble size would be needed to ensure that the empirically calculated rejection rate distributions do not vary too much with different sizes or samples (i.e., the box plots in Figure 1). My feeling is that this aspect of the approach needs more vigorous treatment as, at present, I would be hesitant to apply this method in practice. The authors do state that the choice of ensemble member, subsamples, and subsample members was "rather arbitrary" (line 209), but claim that they are "quite confident" that different choices will not significantly affect the behavior. I confess that I am skeptical of this claim and think that a more thorough evaluation of these parameter choices is needed to demonstrate robustness.

We agree with the reviewer, that there is little evidence in the paper regarding the robustness of the approach under consideration of sizes of ensembles and subsamples. We have done a more systematic evaluation regarding the sizes (# ensemble members, # subsample members, # subsamples) and will include this into the manuscript with a discussion. As expected by the reviewer, our results show that larger ensemble and subsample sizes lead to a higher sensitivity with higher rejection rates and a longer rejection. In the paper we used 50 ensemble members, in Fig. 9 up to 200 ensemble members. This suggests that the methodology can be made even more sensitive at the cost of more computational resources. In order to get a better feeling for this, it would probably make sense to compare the sensitivity, robustness, and computational cost of this methodology to other existing methodologies, but in our opinion this should be part of a greater intercomparison project.

[Figure]

*Figure 9: Rejection rates and decisions for 500 hPa geopotential for the experiment with additional diffusion (D = 0.01) as in Fig. 3 of the paper with different numbers of ensemble members, subsample members, and subsamples. While the difference between 500 (left) and 100 (right) subsamples seems to be negligible, a larger number of ensemble and subsample members clearly increases the sensitivity of the methodology.*

A second concern that I have is in regards to which variables to evaluate in practice. While the authors state that evaluating all of the variables would be "overkill", it is unclear to me how one would best determine the "key" set of variables to use for a particular application. The authors do recognize that "for a fully-coupled GCM, some further considerations will be needed", but it is unclear to me how they chose the key variables even in their limited-area tests (and how they know whether the set of variables they choose was sufficient).

We agree with the reviewer, that this questions is not answered sufficiently in the manuscript. A complete test would require all possible output variables on all grid points on all possible vertical levels, which we think is currently unfeasible for any verification methodology. In the meantime, we have conducted additional experiments with a larger set of variables. In this case (explicit diffusion with a diffusion coefficient of 0.005, see Fig.10), we see a rejection for all tested variables during the first few hours, which can be explained by the many interactions happening in our atmosphere. However, the problem of selecting the key variables which most likely represent most of the model state remains a challenge, especially as the sensitivity of a variable is probably highly

dependent on the actual change in the model. For future work, we consider performing principal component analysis as in Baker et al. 2015. We will update the manuscript with the figure below and discuss this more extensively.

The reviewer is also inquiring regarding application to a fully-coupled GCM. We think that for a fully coupled GCM (assuming ocean, land-surface and sea-ice components) the testing should involve two steps. In a first step, the different components should be verified individually, possibly using different integration periods. In a second step the coupled system would need to be verified, with a particular eye on the variables that are heavily affected by the coupling (e.g. near-surface temperature for ocean-atmosphere coupling). However, further studies of this topic would be needed.

[Figure]

*Figure 10: Global decisions for several variables for an ensemble with little additional explicit diffusion (D = 0.005). The rejection during the first few hours for all output variables shows that the system is highly interactive. This indicates, that instead of looking at all possible output variables at each level, a set of variables covering dynamics, microphysics, and radiation will most likely suffice for finding significant differences in the model state.*

An interesting aspect of this approach is the focus on evaluating the detectability of changes over time (several hours to several months). Looking at the rejection rates over time (e.g., Figures 3-8) allows for comparing the relative importance of the change being tested to the model's internal variability. Perhaps this approach gives insights that other methods may miss (though this is not demonstrated). Many (or most?) of the presented results show that modifications change from "rejected" to "not rejected" as time increases. While I quite like these plots, it is not clear to me in general how this should be interpreted (in practice) in terms of deciding whether the modification should be rejected or not. This change from "reject" to "not reject" over time really highlights the question of "what is this test for?". For example, there is a discussion in Milroy et al. (2018) about how changing a random number generator is a detectable change after several time steps, but not after a year. In that case, Milroy says that this type of change is unimportant to them as the overall climate statistics are consistent at a year. In this paper, I am unclear as to whether the authors are purposely trying to find changes that are rejected at early time steps even if later on in the simulation the changes are not rejected. Is this important to their evaluation? They say that the approach is particularly sensitive to small changes, but is this a good quality? And if so, why? Perhaps the answer is subjective depending on the variable or change being tested, but I believe this point needs to be clarified for the approach to be useful in practice. Particularly because, in my opinion, a strong motivation for statistical ensemble approaches is to remove as much of the

subjectivity (i.e.,need for a climate expert) as possible from the decision on correctness when evaluating model changes.

By looking at the temporal evolution of the rejection rate, we are able to give an indication of the significance of the differences under consideration of the increasing internal variability with time and thus give a recommendation whether a change might be significant for short-term- or long-term-simulations. In the case of automatic testing for continuous integration, we believe that even a difference that is only detected at the very beginning of the simulation should lead to a rejection, as the models are indeed no longer statistically indistinguishable in this case, and we can therefore objectively reject the global null hypothesis. However, we believe that this kind of evaluation is also a reasonable help for (subjective) decisions such as whether the switch to single precision is justified for longer climate simulations. We could of course think of some objective criteria for such decisions, but as such criteria are also always highly subjective, we do not believe that we can get rid of any subjectivity in such cases. Nevertheless, we agree that this might need some clarification in the manuscript and will therefore adapt it.
* * *
specific suggestions/questions:
* * *
-Abstract: line 15: I wouldn't categorize the switch from double-precision to single-precision as "tiny". Maybe this is not what was meant – if not, then please re-phrase or clarify .

We have re-ordered the sentences and changed "tiny" to "very small" in order to make it more clear.

-line 76: For Rosinski and Williams work, consider mentioning what model this was for and why this approach is no longer appropriate on the current model(s).

We will mention the model and might address the excessive growth of perturbation in parameterizations.

-line 103: Note that the POP CESM consistency test concept is almost entirely different as it does not use PCA - but looks at individual variables and at individual grid points (as mentioned) accounting for their location-specific variability.

We will re-formulate the sentence and mention that it does not use PCA. Thank you!

-line 213: define "floored variables"

Thank you. We will add a definition instead of just bringing precipitation as an example.

-last paragraph of 3.1: Will this method catch something that the other methods wouldn't?

We think that the focus on a grid cell level instead of spatial averages could catch some differences that other methods would not (see for example Fig. 11). But of course, without comparing the different methods on the same datasets, we cannot say for sure.

-lines 248-249: Do you perturb all of those variables at the same time?

Yes, we perturb all these variables at the initial model state.

-line 263: Given the (somewhat large) choice of 1e-4, it would be helpful to show the ensemble spread over the first few hours as done in the Milroy 2018 paper that is referred to or somehow give more rigorous justification for this choice.

We agree, that the perturbation is quite strong, but it gives a good spread already at the initial model state and also allows us to apply it to the single precision model without a big influence of round-off errors (i.e. the same perturbation for single precision and double precision). The methodology of course also works with smaller perturbations, but the rejection rate at the initial state (timestep 0) is typically much smaller than after for example one hour. The good ensemble spread at the initial model state for example allows us to identify differences in the diagnostics of the 500 hPa geopotential between single and double precision in Fig. 3. We will try to clarify this in the manuscript.

-Figure 1: This is a nice graphic. My only suggestion would be to add something to indicate that the control and reference ensembles are from the same model. (This is mentioned in the caption, but might be a nice addition to the figure.)

Thank you for the good suggestion! We will adapt the figure.

-line 484: how many variables are there in total?

In total, COSMO possesses 298 output variables. However, many of these are probably not a good indicator for the model state or even constant over time (e.g. surface height). So far, we have not done a detailed analysis regarding how many variables have shown a rejection in this case.
* * *
minor items:
* * *
-line 4 (abstract): "unsuspicious" sounds awkward - maybe "innocuous" ?

-line 26: "are there" => "are intended"

-line 94: "from which many show high correlations with each other" => " many of which are highly correlated"

-line 118: "sensitivity" => "sensitive"

- line 150: suggest putting ()s around equation numbers to match equation label : "equation 1" => "equation (1)" (other lines in the paper as well)

-line 173: replace "went for" with more formal language

-line 182: "used local statistics" => "chosen local statistics" or "selected local statistics" ("used" sounds like it has been tried out already - like as in "used car")

-line 208: same as above (replace "used")

-line 232: "rejection" => "rejections"

-line 364: "Section" is misspelled

-lots of places: "floating point" => "floating-point" when used as an adjective

-line 395: "much less" => "fewer"

-line 444: "the the"

-line 475: (twice): "casted" => "cast"

-line 512: "be effective at" => "affect"

-Throughout: consider an editing pass - especially for missing commas

Many thanks for pointing out these typos/mistakes. We will change them in the manuscript accordingly and also try to improve the language, especially regarding missing commas.

**Answers to Reviewer Comments 2:**

We thank the reviewer for reviewing the paper and the good comments and suggestions. We have addressed the specific comments below.

In the current form, this study reads like an exploratory study with a lack of clear framework for automated testing - which seems to be the goal - and thus seems incomplete. For example, the authors do not prescribe which variables to evaluate, how many variables to use or how long the tests should be run or how many ensembles. While the authors hint at these in text in places, these aspects still lack clear answers. I think it will be best to do additional wider case studies - like those done by others and finalize the framework based on all the results rather than leaving it out for the future.

It is true that the paper has primarily explored a novel verification methodology using one specific model set up. An automated testing framework will in general depend on the model considered (e.g. global version regional, coupled versus atmosphere-only, etc), and cannot fully be answered by our study. However, we have conducted additional simulations where we evaluated more variables and use different ensemble and subsample sizes in order to give a more complete picture of this aspect. The results show some sensitivity to ensemble and subsample size and also to the specific variables (see answers to reviewer comments 1 above). In order to make the methodology operational, we might need to perform more tests. However, the purpose of the work is not to present an operational verification framework, but to present the methodology and provide some evaluation of its versatility. As there will be always a tradeoff between sensitivity, robustness, and computational resources, it is difficult to give a general recommendation for a set of parameters which would be valid for all the specific use cases. However, we think that it would be highly interesting to perform an intercomparison project between existing verification methodologies where we could get a better feeling for these properties, but this would probably merit its own paper.

Also, there is little novelty in the work. While the authors evaluate the null hypothesis at each grid point for the atmosphere - which is a little different from the Baker et al. (2015), Milroy et al. (2018), Mahajan et al. (2017, 2019) and Massonnet et al. (2020) tests for the atmosphere models - the need for doing that is not clear and has not been explored in this study. Atmospheric mean flow fields are highly homogeneous with longer correlation length scales. Fig. 2 is a good example of this which shows high spatial correlation of the 500hPa geopotential height. It may thus be important to argue for the need for this grid point based test more strongly. The authors say that it is more fine-grained and thus would help with debugging. I am not sure how looking at some grid points failing the test would help with debugging. I think a clear case needs to be made, if possible with examples/case studies. If not, I think a comparison with tests that use the domain averages (Baker et al. 2015, etc.) would help justify the need for these fine-grained tests.

We do not agree with the reviewer, that there is little novelty in the work. We think that each of the following points represents a novelty in the field of verification of weather and climate models:

- Evaluation of the atmospheric field on each grid point instead of looking at domain averages. We believe that this approach has significant advantages in some cases. In order to

show this, we have conducted additional experiments where we were varying the tile sizes of the output variables for the local tests. For example, the figure below shows a clear advantage of evaluating the variable at every grid cell compared to tiles of 16x16 grid points, where the methodology is no longer able to detect a very small amount of explicit horizontal diffusion with a diffusion coefficient of 0.005, most probably because such a little amount of diffusion will not affect spatial averages for many variables.

- The design of the methodology with a reference and control ensemble makes the methodology very robust and not dependent on the chosen local statistical hypothesis test, as shown in the manuscript. Even though the used local statistical hypothesis tests are different in terms of the test statistic, and also the power of the test, the results obtained with our methodology do not change.

- The application and evaluation of the test over time is new in this form and gives a good indication of the significance of the difference when compared to the increasing internal variability with time. These results also indicate that more simulations over a short amount of time (days to weeks) might be preferred to fewer but longer simulations.

We believe that a comparison with existing methodologies would require a significant additional effort which would be beyond the scope of this work. Regardless, we think that a thorough intercomparison study of the existing methodologies would be of high value for the verification community, but that this would probably merit its own paper.

[Figure]

*Figure 11: Rejection rates and decisions as in Fig. 3 for total vertically integrated cloud liquid water for an ensemble with additional explicit diffusion (D = 0.005), where the evaluation was done for different tile sizes. The difference can still be detected for smaller tiles (e.g. 4x4 grid points), but is no longer detectable for large tiles (16x16). In this case, spatial averaging hides the difference between the ensembles. The lower mean rejection rate for smaller tiles can be explained with a higher percentage of zero-value tiles, where the statistical test is not able to reject the local null hypothesis. In order to compensate for the lower number of local hypothesis tests per subsample for the spatially averaged ensemble members, the number of subsamples has been increased accordingly (e.g. 1600 subsamples for the 4x4 tiles instead of 100 subsamples for the 1x1 tiles).*

The main difference from previous testing methodologies is the use of mean rejection rates that are derived from sub-samples of control and evaluation ensembles - essentially conducting an ensemble of tests. Other studies only use one test to make a pass or fail decision. However, other tests, for example Mahajan et al. 2019, do use such an ensemble of tests to detect the false negative rates, which is kind of similar to this approach. This difference should be pointed out more clearly in the paper.

We thank the reviewer for this suggestion. We will try to point this out more clearly in the manuscript.

Also, while the authors conduct several case studies, showing that the tests can catch certain small differences, it is not clear how small these differences really are. I think the authors need to pay more attention to the detection capabilities of the test. It may be good to look at more parameters that are used in other studies to establish the robustness of the testing approach.

We agree with the reviewer, that it is difficult to give a good objective measure of the magnitude of the differences. Of course, this is a universal problem, because if we knew the magnitude of these differences already, such a verification methodology would not be necessary. As stated above, we think that a comparison with existing methodologies is beyond the scope of this work and would be better conducted as part of a greater intercomparison project.

I think this may be a useful alternative test to the existing methods, but it needs to be more formalized in its prescription with supporting results and comparisons with other studies.

Please see our comments above.

Other Specific Comments:

Lines 155-170: Discussion of FDR approach. There are several approaches to FDR. See for example, Ventura et al. (2004). It may be good to cite these different approaches here given the nature of the discussion. Also, Mahajan et al. (2021) recently used the FDR approach for testing statistical reproducibility in an ocean model and found it to be quite sensitive. It is interesting to note that the atmosphere model does not show sensitivity to this approach - although details are not presented here. Nonetheless, It may be good to cite this work here, which appears relevant to this discussion.

We will discuss this in more detail and also cite the corresponding work. We chose to not further discuss the relatively low sensitivity of the FDR approach in our case, because we wanted this manuscript to focus more on the methodology we have actually used.

Lines 190-200: Mahajan et al. (2017 and 2019) also used the Monte Carlo approach that is being used here, i.e. they also use a large control ensemble (100 or so members) to establish the rejection rates. They indeed found that this approach yielded similar results to pooling the ensembles together. The approach of pooling ensembles together is called permutation testing and it may make sense to use the term here for clarity. Also, the line, 'Depending on the difference between the two models ….' seems hand wavy. Please clarify or omit.

Thank you for the suggestion. We will try to clarify this paragraph. As we have not compared the two approaches, we will omit the pointed out line.

Lines 220, that paragraph. The FDR approach does not suffer from this issue of the arbitrariness of the significance level of the local null hypothesis, where the p-value is corrected based on the significance level of the global null hypothesis. This should be discussed here since FDR was discussed earlier in the text.

Thank you, we will mention the FDR approach in this paragraph.

Computational costs for Monte Carlo tests. In a few places, the computational cost of running Monte Carlo approaches is mentioned. Given the current computers with accelerators, I think it is generally a weak argument. For example, conducting Monte Carlo tests for all the 150 variables, say used by Baker et al., should not be much of a computational hindrance.

We agree that the computational costs for testing are insignificant compared to for example running a model. Nevertheless, as the methodology performs an evaluation at every grid point for every few hours, the computational costs are still not negligible. Running the verification methodology with 20 subsample members and 100 subsamples for 124 output steps on a 102 x 99 grid takes roughly 3 minutes per variables when using 10 cores. For 150 variables that would mean roughly 7.5 hours runtime (probably a bit less because of caching and memory effects) on 10 cores, which is not negligible. The implementation for sure has some potential for performance optimization, but we consider it already relatively efficient, as the computationally expensive part has been implemented in C++. However, depending on the intended use case and frequency of the test, it is of course

feasible to perform such a test on all variables. We will try to discuss this in more detail in the manuscript.

It might be important to discuss situations where this test may be more useful than others, particularly those that evaluate longer runs and situations where it may be useful to run these other longer tests.

As mentioned above, we think that the focus on grid cells will offer some advantages in finding differences that are no longer visible in spatial means (see Fig. 11). Based on our results, we think that shorter simulations (days to weeks) are generally sufficient for finding differences. However, we have not compared the methodology to other existing methodologies and therefore can only speculate.

References:

*Mahajan, S, 2021: Ensuring statistical reproducibility of ocean model simulations in the age of hybrid computing. In Proceedings of the Platform for Advanced Scientific Computing Conference (PASC '21). Association for Computing Machinery, New York, NY, USA, Article 1, 1–9. DOI:https://doi.org/10.1145/3468267.3470572*

*Valérie Ventura, Christopher J. Paciorek, and James S. Risbey. 2004. Controlling the Proportion of Falsely Rejected Hypotheses when Conducting Multiple Tests with Climatological Data. Journal of Climate 17, 22 (2004), 4343–4356. https: //doi.org/10.1175/3199.1 arXiv:https://doi.org/10.1175/3199.1*

Thank you for the references. We will include these in the manuscript.

---

## Author Response (AR1)

**General Remarks about the changes made to the manuscript:**

After the useful comments from both reviewers we have decided to substantially change and, in our opinion, improve the manuscript. As there are many changes throughout the manuscript, we list the most important ones here:

- We have performed sensitivity tests with regard to ensemble and subsample sizes. We show that higher numbers of ensemble and subsample members lead to a higher sensitivity in the newly added Section 4.8. Considering these results, we think that the initially used ensemble size (50) and subsample size (20) was probably too low for a high sensitivity.
- We haver applied our methodology to data where we use spatial averaging and show that spatial averaging indeed leads to less sensitivity for changes such as additional explicit diffusion (see Section 4.9).
- We propose a set of key output variables and show results for these variables in a clearer way. We also propose a way to determine an objective threshold of rejections for an automated testing framework (see Section 4.10).
- We have found a mistake in our application of the FDR approach. Without this mistake, the FDR approach does in fact show very similar results as our methodology, even though it seems to be a little bit less sensitive (but computationall less expensive). We discuss this in Section 4.11.

Our answers and corresponding changes according the the reviewer comments are addressed in the next few pages.

**Answers to Reviewer Comments 1:**

Many thanks for reviewing our paper and the constructive comments on how we could improve it. Please find our answers to the specific comments below.

The authors describe an ensemble verification method that uses statistical hypothesis testing to assess the effects of minor model changes. Note that other statistical ensembles tests have been previously proposed (which the authors reference) and are in use for evaluating climate models. These types of statistical evaluations are important as bit-reproducibility is not a practical approach for verification (or correctness checking) given the chaotic nature of climate/weather models as well as modern heterogeneous architectures.

The authors' statistical ensemble approach uses a total of three ensembles: the so-called "control" and "reference" ensembles are both from the old or accepted model, and an "evaluation" ensemble is generated from the new model or configuration that is being evaluated for differences. It is important to note that their statistical tests are applied locally -- meaning at each grid cell. The difficulty here is that the local statistical tests can't be assumed to be independent because the data is very likely spatially correlated. (And climate models have many different variables, many of which will have different spatial correlation properties). Therefore, how does one decide on the global reject criteria? The authors rightly admit that this would be non-trivial to determine for all the variables, and for this reason, they generate the two ensembles from the same (old) model (control and reference) to compare and empirically determine the rejection rate distribution. This rejection rate distribution computed from the two ensembles (which are not different) then becomes the baseline against which the rejection rate from the new (evaluation) ensemble as compared to the reference ensemble can be evaluated (nicely illustrated in Figure 1 in the paper). On the whole, this process seems rather involved, though the authors do a nice job explaining their approach. Results are given for a number of experiments for which the anticipated output is then confirmed by the approach. Unfortunately, the authors do not compare their test to other available ensemble statistical tests. Of course a comparison would be useful as well as interesting (the authors acknowledge this), but perhaps the amount of additional effort required is too high...

We do agree that a comparison with already existing statistical tests using ensemble simulations would be very interesting and useful. However, this would require a significant additional effort and we therefore believe it to be beyond the scope of this work. We think that such an evaluation could for example be part of a greater intercomparison project, where the existing methodologies are compared for different cases.

One aspect of this work that I do find concerning is that I do not have a sense of the robustness of this approach. In particular, with ensemble methods, the size of the ensemble typically directly influences the robustness of the method. In this case, I would expect that a much larger ensemble size would be needed to ensure that the empirically calculated rejection rate distributions do not vary too much with different sizes or samples (i.e., the box plots in Figure 1). My feeling is that this aspect of the approach needs more vigorous treatment as, at present, I would be hesitant to apply this method in practice. The authors do state that the choice of ensemble member, subsamples, and subsample members was "rather arbitrary" (line 209), but claim that they are "quite confident" that different choices will not significantly affect the behavior. I confess that I am skeptical of this claim and think that a more thorough evaluation of these parameter choices is needed to demonstrate robustness.

We thank the reviewer for this comment. We have done a more systematic evaluation regarding the sizes (# ensemble members, # subsample members, # subsamples) and included it in the manuscript (see Section 4.8 and Figures 9 and 10). A higher number of ensemble and subsample members does indeed increase the sensitivity of the test.

A second concern that I have is in regards to which variables to evaluate in practice. While the authors state that evaluating all of the variables would be "overkill", it is unclear to me how one would best determine the "key" set of variables to use for a particular application. The authors do recognize that "for a fully-coupled GCM, some further considerations will be needed", but it is unclear to me how they chose the key variables even in their limited-area tests (and how they know whether the set of variables they choose was sufficient).

We thank the reviewer for this comment and we agree, that this issue was not sufficiently addressed in the first version of the manuscript. We now propose a set of key variables we think reflects the most important processes of an atmospheric model (see for example Fig. 4). The choice is still subjective and most probably not sufficient to detect all possible changes. However, this would require a test of all variables at all grid points and vertical levels, which is not feasible with the current computational resources. Due to many variables being highly correlated and based on our results, we think that testing a few standard output variables on selected vertical levels is already sufficient for all but the tiniest changes.

The reviewer is also inquiring regarding application to a fully-coupled GCM. We think that for a fully coupled GCM (assuming ocean, land-surface and sea-ice components) the testing should involve two steps. In a first step, the different components should be verified individually, possibly using different integration periods. In a second step the coupled system would need to be verified, with a particular eye on the variables that are heavily affected by the coupling (e.g. near-surface temperature for ocean-atmosphere coupling). However, further studies on this topic would be needed. We now have made this more clear in the manuscript (see lines 614 - 618).

An interesting aspect of this approach is the focus on evaluating the detectability of changes over time (several hours to several months). Looking at the rejection rates over time (e.g., Figures 3-8) allows for comparing the relative importance of the change being tested to the model's internal variability. Perhaps this approach gives insights that other methods may miss (though this is not demonstrated). Many (or most?) of the presented results show that modifications change from "rejected" to "not rejected" as time increases. While I quite like these plots, it is not clear to me in general how this should be interpreted (in practice) in terms of deciding whether the modification should be rejected or not. This change from "reject" to "not reject" over time really highlights the question of "what is this test for?". For example, there is a discussion in Milroy et al. (2018) about how changing a random number generator is a detectable change after several time steps, but not after a year. In that case, Milroy says that this type of change is unimportant to them as the overall climate statistics are consistent at a year. In this paper, I am unclear as to whether the authors are purposely trying to find changes that are rejected at early time steps even if later on in the simulation the changes are not rejected. Is this important to their evaluation? They say that the approach is particularly sensitive to small changes, but is this a good quality? And if so, why? Perhaps the answer is subjective depending on the variable or change being tested, but I believe this point needs to be clarified for the approach to be useful in practice. Particularly because, in my opinion, a strong motivation for statistical ensemble approaches is to remove as much of the subjectivity (i.e.,need for a climate expert) as possible from the decision on correctness when evaluating model changes.

By looking at the temporal evolution of the rejection rate, we are able to give an indication of the significance of the differences under consideration of the increasing internal variability with time and thus give a recommendation whether a change might be significant for short-term- or long-term-simulations. In the case of automatic testing for continuous integration, we believe that even a difference that is only detected at the very beginning of the simulation should lead to a rejection, as the models are indeed no longer statistically indistinguishable in this case, and we can therefore objectively reject the global null hypothesis. However, we believe that this kind of evaluation is also a reasonable help for (subjective) decisions such as whether the switch to single precision is justified for longer climate simulations. Nevertheless, we agree that this topic needed clarification in the manuscript and have therefore added Section 4.10 "False positives and determining a threshold for automated testing".
* * *
specific suggestions/questions:
* * *
-Abstract: line 15: I wouldn't categorize the switch from double-precision to single-precision as "tiny". Maybe this is not what was meant – if not, then please re-phrase or clarify .

We have re-ordered the sentences and changed "tiny" to "very small" in order to make it more clear.

-line 76: For Rosinski and Williams work, consider mentioning what model this was for and why this approach is no longer appropriate on the current model(s).

We have added the following two sentences (line 98):

*"The methodology of Rosinski and Williamson (1997) was developed and used for the NCAR Community Climate Model (CCM2). However, the approach is no longer applicable for its current successor, the Community Atmosphere Model (CAM), because the parameterizations are ill-conditioned, which makes small perturbations grow very quickly and exceed the tolerances of rounding error growth within the first few timesteps (Baker et al., 2015)."*

-line 103: Note that the POP CESM consistency test concept is almost entirely different as it does not use PCA - but looks at individual variables and at individual grid points (as mentioned) accounting for their location-specific variability.

Thank you, we have now reformulated the sentence (line 121):

*"Baker et al. (2016) also used z-scores for consistency testing of the Parallel Ocean Program (POP), the ocean model component of the Community Earth System Model (CESM). However, instead of evaluating principal components on spatial averages, as in Baker et al. (2015), they applied the methodology at each grid point for individual variables and stipulated that this local test has to pass for at least 90% of the grid points to have the global test pass."*

-line 213: define "floored variables"

Thank you. Due to many changes in the manuscript, we now only use the expression for precipitation, where we use the definition for this case (no negative precipitation, see line 499).

-last paragraph of 3.1: Will this method catch something that the other methods wouldn't?

We think that the focus on a grid cell level instead of spatial averages could catch some differences that other methods would not. We have added Section 4.9 and Fig. 10, where we show that spatial averaging leads to lower sensitivity. But of course, without comparing the different methods on the same datasets, we cannot say how our methodology compares to other approaches with regard to sensitivity.

-lines 248-249: Do you perturb all of those variables at the same time?

Yes, we perturb all these variables at the initial model state.

-line 263: Given the (somewhat large) choice of 1e-4, it would be helpful to show the ensemble spread over the first few hours as done in the Milroy 2018 paper that is referred to or somehow give more rigorous justification for this choice.

We thank the reviewer for this comment. We have added Appendix A, where we discuss the perturbation strength more extensively. We show that a such a relatively strong perturbation only leads to a slightly higher variance during the first few days than a perturbation at machine precision ($1e{-}16$ ). Next to a higher variance after only one time step (initial state), the relatively strong perturbation allows for a fair comparison between single- and double-precision versions of the model.

-Figure 1: This is a nice graphic. My only suggestion would be to add something to indicate that the control and reference ensembles are from the same model. (This is mentioned in the caption, but might be a nice addition to the figure.)

Thank you for the good suggestion! We have adapted the figure.

-line 484: how many variables are there in total?

In total, COSMO possesses 298 output variables. However, many of these are probably not a good indicator for the model state or even constant over time (e.g., surface height).
* * *
minor items:
* * *
-line 4 (abstract): "unsuspicious" sounds awkward - maybe "innocuous" ?

Great term, thank you!

-line 26: "are there" => "are intended"

Changed as suggested.

-line 94: "from which many show high correlations with each other" => " many of which are highly correlated"

We have changed it to "many highly correlated".

-line 118: "sensitivity" => "sensitive"

Changed as suggested.

- line 150: suggest putting ()s around equation numbers to match equation label : "equation 1" => "equation (1)" (other lines in the paper as well)

We have now added parentheses, thank you.

-line 173: replace "went for" with more formal language

We have removed this sentence in the new version of the manuscript.

-line 182: "used local statistics" => "chosen local statistics" or "selected local statistics" ("used" sounds like it has been tried out already - like as in "used car")

Thank you. We have rewritten the sentence, but now use "chosen" instead of "used":

*"The specific definition of H0 (i,j) will be given later, as it somewhat depends upon the statistical hypothesis test used; see Sect. 3.3."*

-line 208: same as above (replace "used")

We have removed this sentence from the manuscript.

-line 232: "rejection" => "rejections"

We have removed this sentence from the manuscript.

-line 364: "Section" is misspelled

Changed, thank you.

-lots of places: "floating point" => "floating-point" when used as an adjective

Changed, thank you.

-line 395: "much less" => "fewer"

Changed, thank you.

-line 444: "the the"

We have removed one of the thes, thank you.

-line 475: (twice): "casted" => "cast"

Changed, thank you.

-line 512: "be effective at" => "affect"

Changed, thank you.

-Throughout: consider an editing pass - especially for missing commas

Many thanks for pointing out these typos/mistakes. We have performed an editing pass and think that the language (and missing commas) should now be improved.

**Answers to Reviewer Comments 2:**

We thank the reviewer for reviewing the paper and the good comments and suggestions. We have addressed the specific comments below.

In the current form, this study reads like an exploratory study with a lack of clear framework for automated testing - which seems to be the goal - and thus seems incomplete. For example, the authors do not prescribe which variables to evaluate, how many variables to use or how long the tests should be run or how many ensembles. While the authors hint at these in text in places, these aspects still lack clear answers. I think it will be best to do additional wider case studies - like those done by others and finalize the framework based on all the results rather than leaving it out for the future.

It is true that the paper has primarily explored a novel verification methodology using one specific model set up. An automated testing framework will in general depend on the model considered (e.g. global version regional, coupled versus atmosphere-only, etc), and cannot fully be answered by our study. However, we agree with the reviewer that the manuscript is not very specific in this area. Therefore, we have now evaluated more variables and use different ensemble and subsample sizes in order to give a more complete picture of this aspect. The results show some sensitivity to ensemble and subsample size and also to the specific variables (see answers to reviewer comments 1 above). However, the purpose of the work is not to present an operational verification framework, but to present the methodology and provide some evaluation of its versatility. As there will be always a tradeoff between sensitivity, robustness, and computational , it is difficult to give a general recommendation for a set of parameters which would be valid for all the specific use cases. However, we think that it would be highly interesting to perform an intercomparison project between existing verification methodologies where we could get a better feeling for these properties, but this would probably merit its own paper.

Also, there is little novelty in the work. While the authors evaluate the null hypothesis at each grid point for the atmosphere - which is a little different from the Baker et al. (2015), Milroy et al. (2018), Mahajan et al. (2017, 2019) and Massonnet et al. (2020) tests for the atmosphere models - the need for doing that is not clear and has not been explored in this study. Atmospheric mean flow fields are highly homogeneous with longer correlation length scales. Fig. 2 is a good example of this which shows high spatial correlation of the 500hPa geopotential height. It may thus be important to argue for the need for this grid point based test more strongly. The authors say that it is more fine-grained and thus would help with debugging. I am not sure how looking at some grid points failing the test would help with debugging. I think a clear case needs to be made, if possible with examples/case studies. If not, I think a comparison with tests that use the domain averages (Baker et al. 2015, etc.) would help justify the need for these fine-grained tests.

We do not agree with the reviewer, that there is little novelty in the work. However, we agree with the reviewer that some aspects of this needed clarification and further experiments, which we have added. We think that each of the following points represents a novelty in the field of verification of weather and climate models:

- Evaluation of the atmospheric field on each grid point instead of looking at domain averages. We believe that this approach has significant advantages in some cases. We have performed some experiments where we show that spatial averaging leads to a reduction in sensitivity. This is discussed in the new Sect. 4.9 and shown in Fig. 10 in the manuscript.

- The design of the methodology with a reference and control ensemble makes the methodology very robust and not dependent on the chosen local statistical hypothesis test, as shown in the manuscript. Even though the used local statistical hypothesis tests are different in terms of the test statistic, and also the power of the test, the results obtained with our methodology do not change.

- The application and evaluation of the test over time is new in this form and gives a good indication of the significance of the difference when compared to the increasing internal variability with time. These results also indicate that more simulations over a short amount of time (days to weeks) might be preferred to fewer but longer simulations.

We believe that a comparison with existing methodologies would require a significant additional effort which would be beyond the scope of this work. Regardless, we think that a thorough intercomparison study of the existing methodologies would be of high value for the verification community, but that this would probably merit its own paper.

The main difference from previous testing methodologies is the use of mean rejection rates that are derived from sub-samples of control and evaluation ensembles - essentially conducting an ensemble of tests. Other studies only use one test to make a pass or fail decision. However, other tests, for example Mahajan et al. 2019, do use such an ensemble of tests to detect the false negative rates, which is kind of similar to this approach. This difference should be pointed out more clearly in the paper.

We thank the reviewer for this suggestion. We have added the following sentences to the manuscript (line 250):

*"Another difference to most existing verification methodologies is that this methodology calculates the mean rejection rate from the evaluation ensemble and the 0.95 quantile from the control ensemble using subsampling. It thus essentially performs multiple global tests to arrive at a pass or fail decision. Most existing methodologies use only one test with all ensemble members for the pass or fail decision. However, many of them use subsampling to estimate the false positive rate."*

Also, while the authors conduct several case studies, showing that the tests can catch certain small differences, it is not clear how small these differences really are. I think the authors need to pay more attention to the detection capabilities of the test. It may be good to look at more parameters that are used in other studies to establish the robustness of the testing approach.

We agree with the reviewer, that it is difficult to give a good objective measure of the magnitude of the differences. Of course, this is a universal problem, because if we knew the magnitude of these differences already, such a verification methodology would not be necessary. As stated above, we think that a comparison with existing methodologies is beyond the scope of this work and would be better conducted as part of a greater intercomparison project. However, we have added and evaluated some more cases with even less diffusion (D=0.005 and D=0.001) than the previously already very small amount (D=0.01).

I think this may be a useful alternative test to the existing methods, but it needs to be more formalized in its prescription with supporting results and comparisons with other studies.

Please see our comments above.

Other Specific Comments:

Lines 155-170: Discussion of FDR approach. There are several approaches to FDR. See for example, Ventura et al. (2004). It may be good to cite these different approaches here given the nature of the discussion. Also, Mahajan et al. (2021) recently used the FDR approach for testing statistical reproducibility in an ocean model and found it to be quite sensitive. It is interesting to note that the atmosphere model does not show sensitivity to this approach - although details are not presented here. Nonetheless, It may be good to cite this work here, which appears relevant to this discussion.

We thank the reviewer for this comment. In fact, we have found a mistake in our implementation of the FDR approach which explained its low sensitivity. The new results show indeed similar results between the FDR approach and our approach, even though the false positive rate seems to be a bit higher with the FDR approach and thus it might be slightly less sensitive. However, considering its lower computational cost (only one test vs many tests), it might indeed be attractive alternative to this approach. Please see the results and discussion of the FDR approach in the newly added Section 4.11 as well as Fig. 11.

Lines 190-200: Mahajan et al. (2017 and 2019) also used the Monte Carlo approach that is being used here, i.e. they also use a large control ensemble (100 or so members) to establish the rejection rates. They indeed found that this approach yielded similar results to pooling the ensembles together. The approach of pooling ensembles together is called permutation testing and it may make sense to use the term here for clarity. Also, the line, 'Depending on the difference between the two models ....' seems hand wavy. Please clarify or omit.

Thank you for the suggestion. We have clarified this in the paragraph and referred to the corresponding results. The part now reads like this (lines 203-211):

*"An alternative to generating the null distribution from a control ensemble is the use of Monte Carlo permutation testing, where one pools two ensembles (from which one does not know yet whether they come from the same distribution), and then applies the test to randomly drawn subsets from the pooled ensemble. This approach allows bypassing the creation of a control ensemble and therefore save compute time. Strictly speaking, the reference value for the number of rejections then comes from a distributionnot produced by one but by two models. Depending on the difference between the two models, this might lead to slightly different results compared to a case where the reference distribution comes from two identical models. However, Mahajan et al. (2017) and Mahajan et al. (2019) used both approaches and found only minor differences between permutation testing and subsampling from a control ensemble to generate the null distribution."*

Lines 220, that paragraph. The FDR approach does not suffer from this issue of the arbitrariness of the significance level of the local null hypothesis, where the p-value is corrected based on the significance level of the global null hypothesis. This should be discussed here since FDR was discussed earlier in the text.

Thank you, we now have mentioned this in the paragraph (line 235):

*"Based on our experience and the results shown in this work, we consider the comparison of the mean to the 0.95 quantile a reasonable choice, eventhough it is not really based on a confidence interval (unlike, for example, the FDR approach discussed in Sect. 2.2)."*

Computational costs for Monte Carlo tests. In a few places, the computational cost of running Monte Carlo approaches is mentioned. Given the current computers with accelerators, I think it is generally a weak argument. For example, conducting Monte Carlo tests for all the 150  variables, say used by Baker et al., should not be much of a computational hindrance.

We agree that the computational costs for testing are insignificant compared to, for example, running a model. Nevertheless, as the methodology performs an evaluation at every grid point for every few hours, the computational costs are still not negligible. Running the verification methodology with 50 subsample members and 100 subsamples for 41 output steps on a 102 x 99 grid takes roughly 110 seconds per variable when using 10 cores. For 150 variables that would mean roughly 4.5 hours runtime on 10 cores, which is not negligible. The implementation for sure has some potential for performance optimization with regard to efficiency and parallelization, but we consider it already relatively efficient, as the computationally expensive part has been implemented in C++. However, depending on the intended use case and frequency of the test, it is of course feasible to perform such a test on all variables. As the FDR approach uses no subsampling, it indeed seems to be a suitable, less expensive, alternative to our approach even though it seems to show a little lower sensitivity. This would allow us to evaluate more variables within the same compute time.

It might be important to discuss situations where this test may be more useful than others, particularly those that evaluate longer runs and situations where it may be useful to run these other longer tests.

As mentioned above, we think that the focus on grid cells will offer some advantages in finding differences that are no longer visible in spatial means. We discuss this and show some results in the newly added Sect. 4.9. Based on our results, we think that shorter simulations (days to weeks) are generally sufficient for finding differences. However, we have not compared the methodology to other existing methodologies and therefore can only speculate.

References:

*Mahajan, S, 2021: Ensuring statistical reproducibility of ocean model simulations in the age of hybrid computing. In Proceedings of the Platform for Advanced Scientific Computing Conference (PASC '21). Association for Computing Machinery, New York, NY, USA, Article 1, 1–9. DOI:https://doi.org/10.1145/3468267.3470572*

*Valerie Ventura, Christopher J. Paciorek, and James S. Risbey. 2004. Controlling the Proportion of Falsely Rejected Hypotheses when Conducting Multiple Tests with Climatological Data. Journal of Climate 17, 22 (2004), 4343–4356. https: //doi.org/10.1175/3199.1 arXiv:https://doi.org/10.1175/3199.1*

Thank you for the useful references! We have included them in the manuscript.

---

## Referee Report (RR1)

Title: "An Ensemble-Based Statistical Methodology to Detect Differences in
Weather and Climate Model Executables"

Authors: Christian Zeman and Christoph Schär

Overall:
* * *
The authors have responded thoughtfully and adequately to all of my concerns and questions. The new manuscript is a significant improvement. The authors clearly put considerable effort into this revision. The additional figures and subsections are quite helpful, and the overall writing quality has improved.

A few notes:
* * *
(1) Figure 4: I quite like this addition that allows the easy comparison across variables
(also Figure 11).

(2) I like that there are 16 "key variables" now being investigated, as in Figure 4.  The variable choice was explained in the reviewer response (e.g., "We now propose a set of key variables we think reflects the most important processes of an atmospheric model ...") and also is commented on in the discussion section of the paper (e.g., line 637).  My suggestion is to add a sentence or two when Figure 4 is first introduced (~line 457) that comments on the choice of 16 variables.

(3) Section 4.9 (and Fig.10) : I appreciate the inclusion of the results on spatial averaging.  I think the bars in Fig. 10 are showing the average of 100 randomly drawn samples, and I am wondering if there was more variability in the global rejection rate for the smaller ensemble and subsample sizes (on the right of the x-axis) than for the larger ones on the left (or is it the other way around).  Just curious...

Also I am wondering about the reason why the 2 bottom plot scenerios in Fig 10 (the CPU and false positive ensembles) have less sensitivity to the spatial averaging than the diffusion modifications.  The paper states that this result "is interesting" (line 565), and it may be that the D = .005 case essentially representing the largest perturbation, followed by the smaller D=.001, then the (likely smaller) CPU perturbation, and then the "smallest" (i.e., "no change").  It does seem to make sense that in the presence of little or no perturbation (so little variability), then the effects of different tile sizes would matter very little... I'm not requesting any paper modifications - just thinking about this.

(4) Appendix A:  This is also a nice addition.

Minor:
* * *
(1) line 372: "visibly or easily" => " visibly or be easily"
(2) line 451: "quite much" =>  "quite a bit"

---

## Author Response (AR2)

**Comment to Topical Editor**

Dear Prof. Dr. Christoph Knote,

Thank you very much for being topical editor for this manuscript and for managing the review process in such a straightforward manner. We have published the source code on Zenodo with a specific DOI and refer to it in the manuscript accordingly.

Thank you very much and kind regards,
Christian Zeman

**Answers to Reviewer 1**

Overall:
* * *
The authors have responded thoughtfully and adequately to all of my concerns and questions. The new manuscript is a significant improvement. The authors clearly put considerable effort into this revision. The additional figures and subsections are quite helpful, and the overall writing quality has improved.

We thank the reviewer for the thorough review of the changes and the good comments. We would also like to thank the reviewer again for the very useful initial review comments, which have helped a lot to significantly improve the manuscript.

A few notes:
* * *
(1) Figure 4: I quite like this addition that allows the easy comparison across variables
(also Figure 11).

Thank you.

(2) I like that there are 16 "key variables" now being investigated, as in Figure 4. The variable choice was explained in the reviewer response (e.g., "We now propose a set of key variables we think reflects the most important processes of an atmospheric model ...") and also is commented on in the discussion section of the paper (e.g., line 637). My suggestion is to add a sentence or two when Figure 4 is first introduced (~line 457) that comments on the choice of 16 variables.

Thank you. We have now added the following sentence around line 457:
"We believe that such a set of variables offers a good representation of the most important processes in an atmospheric model (i.e., dynamics, radiation, microphysics, surface fluxes) and, considering the often high correlation between different variables, is therefore likely sufficient to detect all but the tiniest changes in a model."

(3) Section 4.9 (and Fig.10) : I appreciate the inclusion of the results on spatial averaging. I think the bars in Fig. 10 are showing the average of 100 randomly drawn samples, and I am wondering if there was more variability in the global rejection rate for the smaller ensemble and subsample sizes (on the right of the x-axis) than for the larger ones on the left (or is it the other way around). Just curious...
Also I am wondering about the reason why the 2 bottom plot scenerios in Fig 10 (the CPU and false positive ensembles) have less sensitivity to the spatial averaging than the diffusion modifications. The paper states that this result "is interesting" (line 565), and it may be that the D = .005 case essentially representing the largest perturbation, followed by the smaller D=.001, then the (likely smaller) CPU perturbation, and then the "smallest" (i.e., "no change"). It does seem to make sense that in the presence of little or no perturbation (so little variability), then the effects of different tile sizes would matter very little... I'm not requesting any paper modifications - just thinking about this.

While we have not really tested this, we think that the variability of the global rejection rate is likely going to be higher for the smaller ensemble and subsample sizes. Taking Fig. 9 as an example, the rejections are often not that clear for smaller ensemble and subsample sizes. Furthermore, we also think that the ratio between subsample and ensemble size will have an effect on this, as the quantiles are closer to the mean with a bigger ratio (when, for example, comparing 20/50 vs 100/200 vs 150/200 in Fig. 9).

The absence of any sensitivity of the CPU ensemble to spatial averaging is indeed interesting. We think that, in this case, the test is quite clearly not able to detect differences and, therefore, the number of rejections is very similar to the no-change ensemble, where the number of false positives are not affected by spatial averaging. We also think that not all changes will show such a high sensitivity to spatial averaging as diffusion. Some changes that can be detected might show no sensitivity to spatial averaging at all. However, this would have to be investigated further to give a clear answer.

(4) Appendix A: This is also a nice addition.

Thank you for the suggestion to add something like that.

Minor:
* * *
(1) line 372: "visibly or easily" => " visibly or be easily"
(2) line 451: "quite much" => "quite a bit"

Thank you, we have change it in the manuscript accordingly.